# Modelling climate-substrate interactions on soil organic matter decomposition with the Jena Soil Model

Marleen Pallandt[1,2,3], Marion Schrumpf[1], Holger Lange[4], Markus Reichstein[1], Lin Yu[5] and Bernhard Ahrens[1]

1. Max Planck Institute for Biogeochemistry, Jena, Germany
2. International Max Planck Research School (IMPRS) for Global Biogeochemical Cycles, Jena, Germany
3. Department of Physical Geography, Stockholm University, Stockholm, Sweden
4. Norwegian Institute of Bioeconomy Research, Ås, Norway
5. Department of Earth System Sciences, Hamburg University, Hamburg, Germany

*Correspondence to*: Marleen Pallandt (marleen.pallandt@natgeo.su.se)

**Abstract.** Soil organic carbon (SOC) is the largest terrestrial carbon pool, but it is still uncertain how it will respond to climate change. Especially the fate of SOC due to concurrent changes in soil temperature and moisture is uncertain. It is generally accepted that microbially driven SOC decomposition will increase with warming, provided that sufficient soil moisture, and hence enough C substrate, is available for microbial decomposition. We use a mechanistic, microbially explicit SOC decomposition model, the Jena Soil Model (JSM), and focus on the depolymerisation of litter and microbial residues by microbes at different soil depths, and its sensitivities to soil warming and different drought intensities. In a series of model experiments we test the effects of soil warming and droughts on SOC stocks, in combination with different temperature sensitivities ($Q_{10}$ values) for the half-saturation constant $K_m$ ($Q_{10,Km}$) associated with the breakdown of litter or microbial residues. We find that soil warming leads to SOC losses at a timescale of a century, and that these losses are highest in the topsoil compared to the subsoil. Droughts can alleviate the effects of soil warming and reduce SOC losses by posing strong microbial limitation on the depolymerisation rates, and even lead to SOC accumulation, provided unchanged litter inputs. While absolute SOC losses were highest in the topsoil, we found that the temperature and moisture sensitivities of $K_m$ were important drivers for SOC losses in the subsoil – where microbial biomass is low and mineral-associated OC is high. Furthermore, a combination of drought and different $Q_{10,Km}$ values associated with different enzymes for the breakdown of litter or microbial residues, had counteracting effects on the overall SOC balance. In this study, we show that while absolute SOC changes driven by soil warming and drought are highest in the topsoil, SOC in the subsoil is more sensitive to warming and drought due to the intricate interplay between $K_m$, temperature, soil moisture, and mineral-associated SOC.

## 1 Introduction

Soils are an important component of the global carbon (C) cycle as they store large quantities of C (e.g. Crowther et al., 2019; Fan et al., 2020). Soils can act as C sources or sinks, depending on the balance between C inputs and outputs over time (Davidson, 2020; Kirschbaum, 2006). Apart from plant litter inputs, microbial residues are recognised as important precursors for the formation of stable, mineral-associated soil organic carbon (Cotrufo et al., 2013; Kallenbach et al., 2016; Liang et al., 2017; Xiao et al., 2023). Therefore, to determine whether soils are a net C source or sink, the speed at which soil organisms decompose litter inputs and existing soil organic carbon (SOC) stocks including microbial residues is of particular importance (Kallenbach et al., 2016).

Soil temperature and soil moisture are two primary controlling factors of microbial decomposition rates, and thereby the carbon turnover rate of soils (Davidson and Janssens, 2006; Moyano et al., 2013; Yan et al., 2018). Additionally, the interaction between microbial SOC decomposition, and adsorption and desorption of SOC to mineral surfaces is an important controlling factor determining the fate of SOC stocks (Ahrens et al., 2020; Dwivedi et al., 2017; Sokol et al., 2022). Given the importance of SOC stocks and their sensitivities to climate change, a better understanding and representation of these complex interactions in coupled C cycle-climate models is extremely important for a better understanding of the carbon-climate feedback (Todd-Brown et al., 2014).

Temperature and warming effects on SOC stocks and soil respiration have been extensively studied before (Kirschbaum, 2006; Subke and Bahn, 2010; Tang et al., 2019), both experimentally (e.g. Benbi et al., 2014; Bradford et al., 2019; Carey et al., 2016; Chen et al., 2024; Conant et al., 2011; Gentsch et al., 2018; Gillabel et al., 2010; Hao et al., 2023; Hartley et al., 2021; Hicks Pries et al., 2017; Li et al., 2020; Moinet et al., 2020; Ofiti et al., 2021; Qin et al., 2019; Reichstein et al., 2005; Soong et al., 2021; Walker et al., 2018; Wang et al., 2022a; Yan et al., 2022) and with coupled C cycle-climate models (e.g. Bauer et al., 2008; Georgiou et al., 2024; Koven et al., 2017; Todd-Brown et al., 2018; Varney et al., 2020; Zhang et al., 2022). Most of these coupled C cycle-climate models, however, have simple process representation with conceptual SOC pools that decay according to first-order kinetics and rate modifier functions to represent the effects of soil temperature and soil moisture changes, and/or the effects of soil clay content on SOC decay rates (Le Noë et al., 2023; Sierra et al., 2015). Since the paradigm shift that SOC persistence is mediated by microbial activity and organo-mineral interactions (Schmidt et al., 2011), SOC decomposition and stabilisation can be more mechanistically described in models. This was realised by the development of next generation soil models (NGSM) of various complexity (Le Noë et al., 2023; Sulman et al., 2018). These NGSMs have direct representation of microbial biomass which drives depolymerization of SOC pools through enzyme production, and representation of mineral associated organic matter, which both reduce microbial growth and thus depolymerisation by limiting substrate availability (e.g. Abramoff et al., 2017, 2019; Sulman et al., 2014; Tang and Riley, 2015, 2024; Wang et al., 2015; Wieder et al., 2014; Yu et al., 2020; Zhang et al., 2022). Through these mechanistic process representations in NGSMs, more explicit representations of their individual temperature and moisture dependencies can be explored.

According to current process understanding, it is now, for example, possible to assign microbial depolymerisation rates a higher temperature sensitivity than SOC sorption and desorption processes (Ahrens et al., 2020; Tang and Riley, 2015; Wang et al., 2012, 2013). However, of the enzymatic driven depolymerisation steps, which are often described using (reverse) Michaelis-Menten kinetics (Tang, 2015), so far mostly effects of the temperature dependence of the maximum depolymerisation rate ($V_{max}$) were explored (Fanin et al., 2022; Liu et al., 2022; Nottingham et al., 2016; Wang et al., 2012). Several studies showed, however, that also the half-saturation constant, $K_m$, is temperature sensitive (Davidson et al., 2006; Davidson and Janssens, 2006; Wang et al., 2012, 2013). Although the number of laboratory studies on the $Q_{10}$ values for $K_m$ of soil enzymes kinetics is limited, values were shown to range between 0.7 and 2.8 for different enzymes (Allison et al., 2018b; Gershenson et al., 2009; Stone et al., 2012; Wu et al., 2022). When the $Q_{10}$ value for $K_m > 1$, this indicates that the relationship between temperature and the substrate binding affinity of the enzyme ($K_m$) is positive, whereas a $Q_{10}$ values < 1 indicate a negative temperature sensitivity of $K_m$ (Allison et al., 2018b; Tang et al., 2019). In the Michaelis-Menten equation (Michaelis and Menten, 1913), $K_m$ is in the denominator, which means that a positive temperature sensitivity of $K_m$ can reduce the enzymatic reaction rate, or vice-versa, increase the reaction rate when the temperature sensitivity of $K_m$ is negative (Fig. 1; Abramoff et al., 2019; Davidson et al., 2006; Davidson and Janssens, 2006).

At high substrate concentrations, the relative importance of the temperature sensitivity of $K_m$ may be smaller than that of $V_{max}$ (Wang et al., 2012), although this has not been explored much in modelling studies and only for positive temperature

sensitivities of $K_m$ (Davidson et al., 2006). In their study, Davidson et al. (2006) used $Q_{10}$ values of 2 for $V_{max}$ and of 1, 1.5 or 2 for $K_m$, and showed that the individual temperature sensitivities of $K_m$ and $V_{max}$ can cancel each other out at low substrate concentrations. Davidson et al. (2006), however, did not study how dynamic changes in the substrate would interact with

80 temperature sensitivity of $K_m$. Furthermore, the soil enzyme kinetic study by Allison et al. (2018b) showed that the $Q_{10}$ values of $K_m$ for enzymes primarily associated with the breakdown of plant derived litter was generally positive ($Q_{10}$ values $> 1$), whereas those for enzymes associated with the breakdown of microbial residues were negative ($Q_{10}$ values $< 1$). Such opposing temperature sensitivities of $K_m$ for different substrate types have not been explored in dynamic modelling studies before.

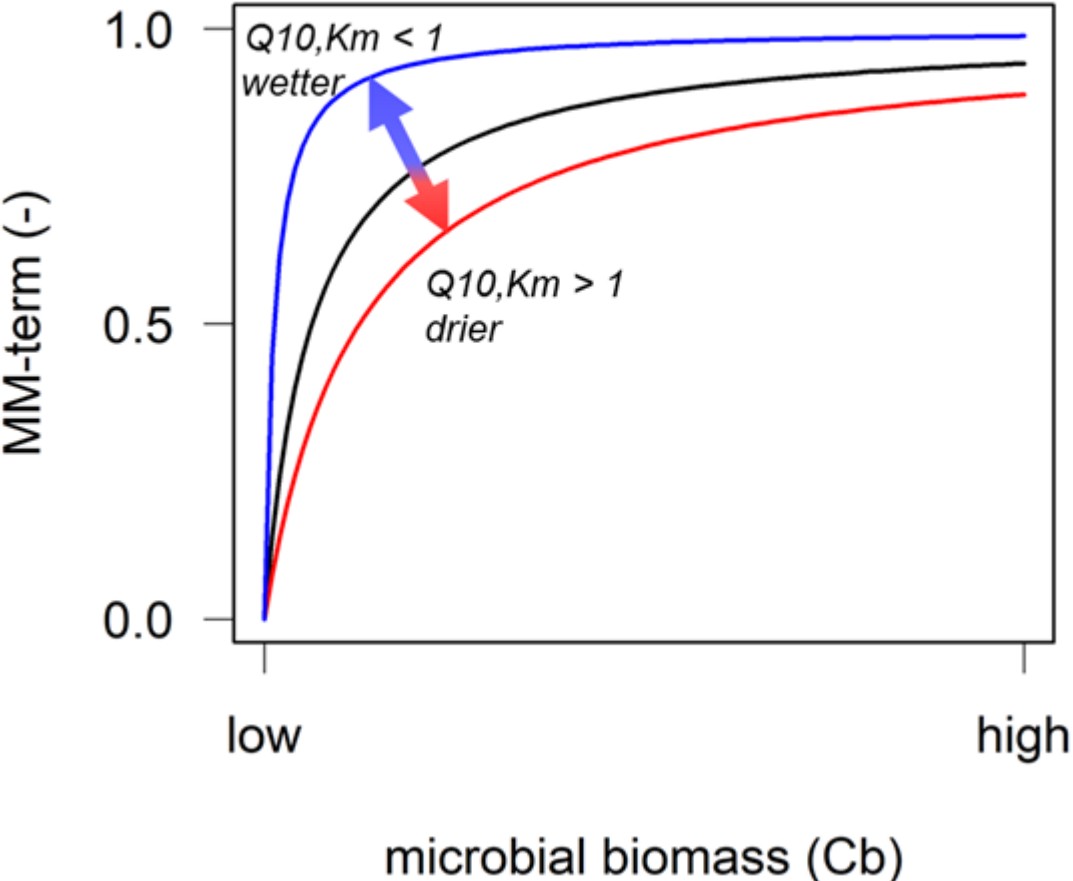

**Figure 1: Conceptual depiction of the relationship between microbial biomass ($C_B$) and a reverse Michaelis-Menten (MM) term, defined as $C_B/(K_m + C_B)$, to represent microbial limitation of the depolymerisation rate. In a forward MM-term (for e.g. microbial uptake of DOC), $C_B$ can be replaced by the substrate concentration. Temperature and soil moisture sensitivities of the half-saturation constant $K_m$ can increase or decrease the MM-term: The temperature sensitivity of $K_m$ can be negative ($Q_{10,Km} < 1$) and thus increase the MM-term (blue line), be absent ($Q_{10,Km} = 1$, black**
**line), or be positive ($Q_{10,Km} > 1$) and thus decrease the MM-term (red line). The MM-term decreases when the soil gets drier or increases when the soil gets wetter (blue line).**

Besides rising soil temperatures, also soil moisture is expected to change in the future with an increased risk of regional droughts (Berg and Sheffield, 2018). Drought can reduce temperature-induced SOC losses by reducing substrate availability
for microbial decomposition (e.g. Schimel, 2018). The moisture sensitivity of the Michaelis-Menten term depends on soil moisture, where lower soil moisture values result in stronger substrate limitation (in the case of forward MM-kinetics) on microbial uptake, or stronger microbial limitation (in the case of reverse MM-kinetics) on enzymatic depolymerisation (Fig. 1; Zhang et al., 2022). At present, the response to soil moisture changes is less well documented and implemented in

microbially explicit SOC decomposition models (Wang et al., 2020), but some have integrated (semi-)mechanistic moisture sensitivity functions to represent substrate and sometimes also oxygen limitation on decomposition rates using volumetric water content (Davidson et al., 2012; Yan et al., 2018) or soil matric potential (Ghezzehei et al., 2019; Manzoni et al., 2014). In some NGSMs, reduced substrate availability directly affects microbial C uptake/growth, and depolymerization rates (Abramoff et al., 2017; Wang et al., 2015; Yu et al., 2020). Not only soil moisture, but also organo-mineral interactions affect substrate availability for microbial depolymerisation because microbes and soil minerals compete for the available SOC (Ahrens et al., 2015). In return, soil moisture and temperature affect mineral sorption and desorption rates. In NGSMs that are microbially explicit and include organo-mineral interactions, these temperature and moisture sensitivities can now also be specifically assigned to each process (Ahrens et al., 2020) and further explored in the context of climate change.

Albeit at lower concentrations, subsoils store huge amounts of OC (e.g. Blume et al., 2016). Recently, the response of deep soils to climate change has received increased attention (Hicks Pries et al., 2023), especially with regard to soil warming (Chen et al., 2024; Hao et al., 2023; Hicks Pries et al., 2017; Li et al., 2020; Ofiti et al., 2021; Soong et al., 2021) and the role of mineral associated organic matter (Benbi et al., 2014; Gentsch et al., 2018; Georgiou et al., 2024; Gillabel et al., 2010; Hartley et al., 2021; Sokol et al., 2022). However, most NGSM studies assumed so far that the soil is only composed of one homogenous layer (Wieder et al., 2015), without considering the vertical profile, with its differences in OC inputs from plant and roots, substrate availability and microbial biomass, as well as associated gradients in organo-mineral associations, temperature and moisture (Hicks Pries et al., 2023). As the relevance of $K_m$ increases with declining microbial biomass (Fig. 1), these gradients within the soil profile will also affect SOC responses to climate change (Pallandt et al., 2022).

Overall, the climate sensitivities of microbially mediated SOC decomposition and their potential impacts on SOC stocks at different soil depths have been partially overlooked in dynamic modelling studies. In this study, we bridge these gaps by applying the C cycle version of the Jena Soil Model (JSM, Yu et al., 2020) to investigate the dynamic interactions between soil moisture, soil temperature and C substrates at different soil depths. JSM is a vertically resolved NGSM, with mechanistic descriptions of microbially driven decomposition and organo-mineral interactions, so that C substrate depletion by microbes or sorption can be explicitly simulated at different soil depths. We use JSM to test the various soil temperature and soil moisture controls on SOC decomposition either individually or simultaneously. Specifically, we focus on the following research questions: 1) How do temperature and soil moisture changes affect modelled SOC decomposition through $V_{max}$ and the Michaelis-Menten term?; and 2) Do top- and subsoil layers respond differently to warming and drought? In a series of model experiments we test the effects of soil warming and different drought intensities on SOC stocks, in combination with different temperature sensitivities of $K_m$ associated with the depolymerisation of the polymeric litter pool or of microbial residues.

## 2 Methods

### 2.1 Model description

JSM is a vertically explicit soil organic matter (SOM) decomposition model with microbial interactions and representation of organic matter (de)sorption to mineral surfaces. For this study, we represent and describe the C cycle, but JSM is also capable of simulating the coupled C, N and P cycles and isotope ($^{13}$C, $^{14}$C, $^{15}$N) tracking. A full mathematical description of JSM and its coupled nutrient cycles can be found in the supplement material of Yu et al. (2020). Here, we summarise the most important C cycle processes relevant to this study, with a conceptual overview in Fig. 2 and parameters and units listed in Table 1.

The soil profile is divided into multiple layers, which receive vertically distributed root litter inputs, or transported C from other soil layers via advection or bioturbation (Ahrens et al., 2020). Additionally, the top soil layer receives aboveground plant

litter inputs (Fig. 2). Above- and belowground non-woody litter inputs are partitioned into soluble and polymeric litter following Parton et al. (1993). JSM does not explicitly simulate enzyme production, but this is implicitly described using Michaelis-Menten (MM) kinetics. For the depolymerisation steps, reverse Michaelis-Menten kinetics is used, as Tang and Riley (2019) found these to be more appropriate than traditional, forward MM-kinetics. The depolymerisation rate (mmol C $m^{-3}$ $hour^{-1}$) of litter or microbial residues into the dissolved organic carbon (DOC) pool (following Ahrens et al., 2015, 2020) is described as:

$$f_{depoly,X} = V_{max,X} \times f_{V_{max,X}}(T_{soil}) \times C_X \times \frac{C_B}{Km_X \times f_{Km_X}(T_{soil},\theta) + C_B} \tag{1}$$

so that the depolymerisation rate is limited by the size of the microbial biomass pool ($C_B$). $X$ is either the polymeric litter pool ($P$) or the microbial residues pool ($R$), $V_{max,X}$ is the maximum specific depolymerisation rate of $X$, $C_X$ is the respective litter pool $X$, and $C_B$ is the microbial biomass pool. $f_{V_{max,X}}(T_{soil})$ is an exponential function expressed with a $Q_{10}$ base (Wang et al., 2012):

$$f_{V_{max,X}}(T_{soil}) = Q_{10,Vmax,X}^{\frac{T_{soil}-T_{ref}}{10}} \tag{2}$$

where $Q_{10,Vmax,X}$ is the temperature sensitivity of the maximum specific depolymerisation rate of litter pool $X$, and $T_{soil}$ and $T_{ref}$ are the soil temperature and reference temperature, respectively.

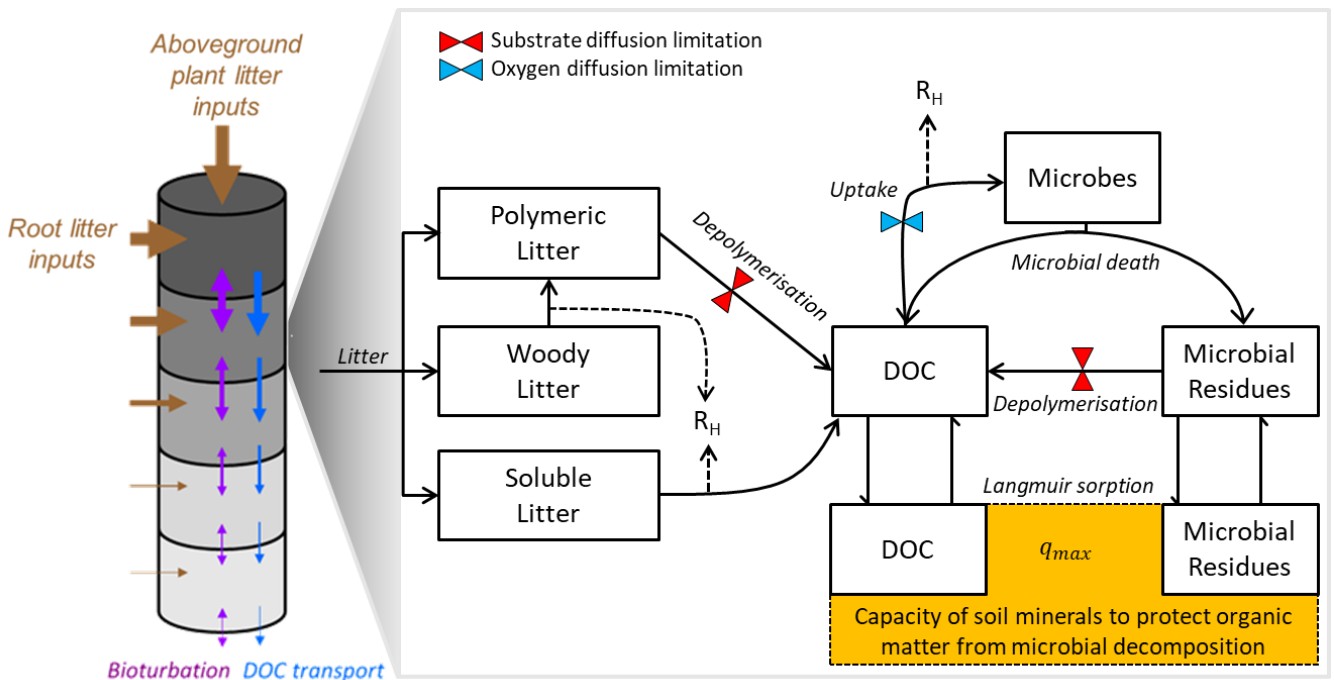

**Figure 2: Schematic representation of the C cycle in JSM, after Yu et al. (2020) and Ahrens et al. (2020). Each soil layer (grey cylinders) receives vertically resolved C litter inputs (brown arrows) from roots, and aboveground litter enters the first soil layer. Carbon can be transported between soil layers through bioturbation (purple arrows) and downwards with the water flux through advection (blue arrows). Carbon pools (black rectangles) and C fluxes (black arrows) are considered for each individual soil depth: Litter inputs are partitioned into a polymeric, woody and soluble litter pool. Polymeric litter and microbial residues can be depolymerised to DOC (Dissolved Organic Carbon) by microbes. DOC and microbial residues can adsorb to mineral surfaces to form MAOM (Mineral-Associated Organic Matter, yellow box) and desorb through Langmuir sorption, where $q_{max}$ is the maximum sorption capacity for sorption of DOC and microbial residues to mineral surfaces. The dotted lines are heterotrophic respiration ($R_H$) fluxes. The coloured hourglasses represent different soil moisture controls on SOC decomposition steps: Microbial limitation of depolymerisation (red), and oxygen limitation (blue) on microbial C uptake for growth.**

**Table 1: Parameter values and constants related to temperature sensitive processes in JSM.**

| Parameter/constant | Value | Unit | Reference |
|---|---|---|---|
| $V_{max,P}$ | 0.1849 | yr$^{-1}$ | Yu et al. (2020) |
| $V_{max,R}$ | 0.2317 | yr$^{-1}$ | Yu et al. (2020) |
| $V_{max,U}$ | 95.76 | day$^{-1}$ | Yu et al. (2020) |
| $K_{m,P}$ and $K_{m,R}$ | 3.70 | mmol C m$^{-3}$ | Yu et al. (2020) |
| $K_{m,U}$ | 85.26 | mol C m$^{-3}$ | Yu et al. (2020) |
| $K_{m,O2}$ | 0.001 | – | Yu et al. (2020) |
| $Q_{10,Vmax.U}$ | 1.98 | – | Allison et al. (2010) |
| $Q_{10,Vmax,P}$ and $Q_{10,Vmax,R}$ | 2.16 | – | Wang et al. (2012) |
| $Q_{10,Km.P}$ | 1.31* | – | Allison et al. (2018b) |
| $Q_{10,Km.R}$ | 0.7* | – | Allison et al. (2018b) |
| $Q_{10,adsorption}$ | 1.08 | – | Wang et al. (2013) |
| $Q_{10,desorption}$ | 1.34 | – | Wang et al. (2013) |
| $T_{ref}$ | 293.15 | K | Wang et al. (2012) |

All $Q_{10}$ values are reported for the model's reference temperature ($T_{ref}$) of 293.15 K (20 °C), and their respective activation energies were taken from literature (Allison et al., 2010; Wang et al., 2012, 2013) by Ahrens et al. (2020, Table 1). $Q_{10}$ values marked with * are unique to this study and taken from Allison et al. (2018b), and measured at a reference temperature of 289.15 K (16 °C). Activation energies for JSM were adjusted accordingly using Eq. (3). All units in JSM are in SI units (mol, seconds and Kelvin), but for table readability and direct cross-referencing values are reported directly from the literature.

The $Q_{10}$ coefficient is the ratio of reaction rates when temperature increases by 10 K. For use within JSM, all $Q_{10}$ values were converted to apparent activation energies ($E_a$) for the model's reference temperature ($T_{ref}$) of 293.15 K following Eq. (7) from Wang et al. (2012):

$$Q_{10} = exp\left[\frac{Ea}{R \times T_{ref}} \times \frac{10}{T_{soil}}\right] \tag{3}$$

where $R$ is the universal gas constant, and $T_{soil}$ is the soil temperature. Inclusion of soil moisture is done through Eq. (4):

$$f_{Km,X}\ (T_{soil}, \theta) = Q_{10,Km,X}^{\frac{T_{soil} - T_{ref}}{10}} \times \left(\frac{\theta}{\theta_{fc}}\right)^{-3} \tag{4}$$

This is a function to describe the sensitivity of the half-saturation constant ($Km_X$) to soil moisture and temperature (Davidson et al., 2012), where $Q_{10,Km,X}$ is the temperature sensitivity of the half-saturation constant of $C_X$, and where $\theta$ and $\theta_{fc}$ are the volumetric water content and water content at field capacity, respectively.

Microbial C uptake for growth is described using traditional, forward MM-kinetics (Tang and Riley, 2019):

$$f_{Uptake} = V_{max,U} \times f_{V_{max,X}}\ (T_{soil}) \times C_B \times \frac{C_{DOC}}{K_{m,U} + C_{DOC}} \times \frac{a^{4/3}}{K_{m,O2} + a^{4/3}} \tag{5}$$

so that the uptake rate (mmol C m$^{-3}$ hour$^{-1}$) is limited by the size of the available substrate ($C_{DOC}$) and by the air-filled pore space ($a$), which is calculated as

$$a = \frac{\theta_{fc} - \theta}{\theta_{fc}} \tag{6}$$

and functions as a proxy to describe the amount of oxygen available for the reaction (Davidson et al., 2012). $V_{max,U}$ is the maximum uptake rate of DOC by $C_B$, $C_{DOC}$ is the dissolved organic C pool, $K_{m,U}$ is the half-saturation constant for the uptake of DOC by $C_B$ and $K_{m,O2}$ is the half-saturation constant of the reaction with oxygen. DOC and microbial residues can adsorb to mineral surfaces to form MAOC (Mineral-Associated Organic C), which is protected from microbial decomposition (Fig. 2, yellow box). In JSM, adsorption and desorption rates are temperature dependent (with literature based $Q_{10}$ values, reported in Table 1), with a full description of the process implementation and successful application in Ahrens et al. (2020).The Particulate Organic C (POC) pool, accessible to microbes for depolymerisation and growth, consists of polymeric litter, DOC and microbial residues (Fig. 2). Soil moisture, water fluxes between soil layers, as well as the maximum sorption capacity ($q_{max}$) are affected by soil texture (described in Ahrens et al., 2015, 2020; Thum et al., 2019; and Section 2.2 Yu et al., 2020).

## 2.2 Vertical process representation in the Jena Soil Model

JSM requires depth-specific soil temperature, soil moisture and litterfall forcing data at a half hourly time step as input. Following Thum et al. (2019), these soil forcing data were generated for a temperate forest site in Germany (Hainich, DE-Hai): First, by running the QUINCY land surface model for 500 years beforehand to bring the soil C pools into equilibrium, using site-specific information on soil physical and chemical parameters (soil texture, bulk density, pH), plant functional type, rooting depth, and meteorological reanalysis data from 1901 - 1930. Second, directly following the 500-year spinup period, QUINCY was run as a transient simulation in combination with FLUXNET3 forcing data for Hainich from 1901 - 2012. These site-specific soil forcing data are then used for JSM model spinup for 500 years to bring the soil C pools in equilibrium, where soil forcing data from 2000 - 2012 are used repeatedly. The soil moisture and soil temperature forcing data from 2000 - 2012 are referred to as ambient soil moisture and soil temperature data from this point. After the 500-year spinup period, JSM is run for 100 simulation years for each model experiment (Section 2.4), again using the soil forcing data from 2000 - 20212 repeatedly.

JSM works with 15 soil layers up to 9.5m depth, where soil layer thickness increases with increasing soil depth (Yu et al. 2020). In this study we focus on the first 6 soil layers between 0 and 50 cm depth, at the following depth intervals: 0 – 6, 6 – 13, 13 – 20, 20 – 26, 26 – 36, and 36 – 50 cm. In each individual soil layer, all C pools and fluxes, as well as the C fluxes between the different layers are tracked through time (Fig. 2, Eqs. 1 – 6). Through its vertically explicit structure and input data, the model is capable of considering processes and variables that vary with soil depth, such as soil moisture, soil temperature, root litter inputs, SOC content, microbial biomass and the sorption and desorption of DOC and microbial residues to mineral surfaces. In the topsoil layers, microbial biomass is highest and closely follows the distribution of the root litter inputs (Ahrens et al., 2020) generated by the QUINCY model. In the deeper soil layers, the fraction of C adsorbed to mineral surfaces (MAOM) increases, and advective and diffusive transport C inputs increase with depth (Ahrens et al., 2020).

## 2.3 Choice of $Q_{10,Km}$ values for polymeric litter and microbial residues

Microbes process SOC by depolymerising a wide array of C substrates derived from plant litter or microbial residues, which greatly differ in their chemistry (Buckeridge et al., 2022; Cotrufo and Lavallee, 2022), and therefore produce a variety of different enzymes that target those different substrate types. In JSM, the polymeric litter and microbial residues pool are depolymerised by extracellular enzymes produced by the microbial pool ($C_B$) to enter the DOC pool (Fig. 2, Eq. (1)). Enzyme production is not explicitly simulated, but assumed to be proportional to the size of the microbial biomass pool. The half-

saturation constants for depolymerisation of polymeric litter and microbial residues ($K_{m,X}$, Eq. (1)) are temperature dependent, but knowledge about their value is restricted to laboratory studies of individual enzymes. In this study, we explore different temperature sensitivities, expressed as $Q_{10}$ values, for our model's half-saturation constants for microbial depolymerisation of polymeric litter ($K_{m,P}$) and microbial residues ($K_{m,R}$). We base these $Q_{10}$ values on the study from Allison et al. (2018b), who give an extensive overview of the temperature sensitivities of different enzymes and their substrate targets. We chose values from this study that would likely be, or closely resemble, the main enzymes involved in the breakdown of our model's polymeric litter ($C_P$) and microbial residue ($C_R$) pools. For the depolymerisation of $C_P$ we targeted a value measured for the enzymes β-xylosidase and total oxidase, as these are involved in the degradation of hemicellulose, lignin and phenolics. For the depolymerisation of $C_R$ we selected a $Q_{10,Km,R}$ value measured for the enzyme leucine aminopeptidase, which is involved in the degradation of polypeptides, the main component of microbial cell walls. The selected value for the depolymerisation of polymeric litter $Q_{10,Km,P}$ is 1.3 (Table 2), i.e. a positive temperature sensitivity of $K_m$, and that for the depolymerisation of microbial residues $Q_{10,Km,R}$ is 0.7, i.e. a negative temperature sensitivity of $K_m$. In the various model experiments (described in more detail in section 2.4), we explore the effects of these different temperature sensitivities on SOC decomposition. Because so little is known about these effects through the temperature sensitivity of the Michaelis-Menten term, we conduct experiments where we assign individual $Q_{10,Km}$ values for the depolymerisation of litter and microbial residues, but also explore the effect of using only a negative temperature sensitivity of $K_m$ (representing a hypothetical situation where all available C substrates have a $Q_{10,Km}$ value < 1), or only a positive temperature sensitivity of $K_m$ (representing a hypothetical situation where all available C substrates have a $Q_{10,Km}$ value > 1).

## 2.4 Model experiments

After the 500-year spinup period (Section 2.2) we conducted an ambient model run, where the 13 years of half-hourly ambient soil moisture and soil temperature forcing data (generated by the QUINCY model, as described in Section 2.2) are recycled for a 100-year simulation. Simple linear regression on the slope of the change in SOC pools was used to determine whether the SOC pools between 0 - 50 cm depth increased or decreased significantly over the 100-year simulation period, i.e. to verify that the SOC pools reached steady state after the 500-year spinup period. Then, to investigate the effects of soil warming on SOC decomposition, we ran several warming experiments in which all ambient soil temperatures were increased by 4.5 K throughout the whole soil column over the 100-year simulation period, keeping the original seasonality in the ambient input data intact and without altering the ambient soil moisture (SM) values. We chose a 4.5 K step increase for soil warming, because soils, including the deep soil up to 1m, are expected to warm by 4.5 K by the end of the century under representative concentration pathway (RCP) 8.5 (Soong et al., 2020). To test the sensitivity of SOC decomposition to warming and to investigate the potential feedbacks through the temperature sensitivity of the half-saturation constants in the Michaelis-Menten term for microbial depolymerisation (Eq. 1), we ran four warming experiments using different values for $Q_{10,Km,P}$ and $Q_{10,Km,R}$ (Table 1): 1) Both $Q_{10,Km,P}$ and $Q_{10,Km,R}$ values are 1 (i.e. not temperature sensitive); 2) Separate $Q_{10}$ values for the breakdown of the microbial residue pool and the polymeric litter pool, where $Q_{10,Km,R}$ is set to 0.7 and $Q_{10,Km,P}$ is set to 1.3 (Section 2.3). Then, to explore the effects of only negative or positive $Q_{10,Km}$ values for depolymerisation (i.e. the effect of having different C substrate compositions), we ran warming experiments 3) Both $Q_{10,Km,P}$ and $Q_{10,Km,R}$ are 0.7 (representing a hypothetical situation where all available C substrates would have a $Q_{10,Km}$ value below 1); and 4) Both $Q_{10,Km,P}$ and $Q_{10,Km,R}$ are 1.3 (representing a hypothetical situation where all available C substrates would have a $Q_{10,Km}$ value above 1). All model experiment settings are summarised in Table 2.

Then, to investigate the effects of soil warming and drying on SOC decomposition, we ran the first set of drought experiments, where we keep all $Q_{10,Km,X}$ values at 1 and use ambient soil temperature + 4.5 K. Soil drying is expected for most of the globe (Wang et al., 2022b and references therein), but drought intensity is uncertain and may vary locally (Cook et al., 2020; Hsu and Dirmeyer, 2023). Soil moisture change projections are very uncertain: 60 projected global lateral and vertical distributions of future soil moisture were highly diverse both in their predicted spatial and vertical distributions (Berg et al., 2017). The multi-model mean of this study showed reductions in subsurface and deep surface soil moisture up to 30% by the year 2100. Given the large divergence between these model projections, we chose to simulate drought by reducing the depth-specific soil moisture values from the forcing dataset in steps of 10%, to be able to compare the effects of a relatively mild versus increasingly stronger droughts on SOC stocks. Specifically, we compared three drought scenarios, where the model's ambient SM inputs are reduced by 10%: Each ambient SM value is multiplied by 0.9, 0.8 or 0.7, respectively (Table 2). As with the warming experiment, the original seasonality in the ambient SM input values is kept intact.

As a last step, we investigate the combined effects of soil warming and drying on SOC decomposition including the feedback through the half-saturation constants' temperature sensitivities. Similar to the first set of 3 drought experiments, ambient soil temperature is raised by 4.5 K, and three different drought intensities are simulated (SM * 0.9, SM * 0.8 and SM * 0.7). Reflecting the most likely realistic combination of $Q_{10,Km}$ values for microbial depolymerisation, as soil will contain both microbial residues as well as polymeric litter for microbes to depolymerise, we used the two individual $Q_{10,Km}$ values from Allison et al. (2018b) for the breakdown of the microbial residue pool and the polymeric litter pool ($Q_{10,Km,R} = 0.7$ and $Q_{10,Km,P} = 1.3$).

**2.5 Model output analyses**

Each model experiment was run for 100 simulation years, yielding daily output files for different soil variables. We calculate SOC stocks as the sum of the soluble litter, polymeric litter, DOC, microbial residues, adsorbed DOC and adsorbed microbial residues pools (Fig. 2). Woody litter is excluded as it is considered part of the aboveground litter layer. To calculate the annual changes in SOC stocks, expressed as percentage change (%) since the start of the simulation, we used SOC values from the last day of each simulation year. All analyses and plots were done using packages "tidyverse", "ggplot2" and "viridis" under R version 4.3.1 in Rstudio (Garnier et al., 2023; R Core Team, 2023; RStudio Team, 2018; Wickham, 2016; Wickham et al., 2019). In Figs. 3 – 5 and A1 – A2, solely as a visual aid, a smoothed line is added to the annual data points using ggplot2 function geom_smooth().

**3 Results**

**3.1 Warming effects on SOC decomposition**

**3.1.1 Modelled SOC stock changes at ambient and elevated soil temperatures**

To check whether JSM reached steady state after spinup, one model run was continued with ambient soil temperatures and soil moisture, and with $Q_{10,Km,X}$ values of 1 (not temperature sensitive), i.e., the same setup as during spinup. A linear regression test confirmed that the first 6 soil layers (0 - 50 cm) are in steady state at Hainich forest, as there is no SOC loss or accumulation over the complete simulation period (Fig. 3, dark blue). The small interannual variability in modelled SOC stocks reflects the interannual variability in the litter inputs and other forcing. Warming the soil by 4.5 K in a model experiment leads to SOC losses for all simulation years (Fig. 3, purple) until 5.1% of initial stocks are lost by the end of the simulation period (Table 2). The topsoil loses more SOC (- 6.2%) than the subsoil (- 3.9%), which is related to the fact that mineral-associated organic C

(MAOC) increases with depth which leaves less available substrates for microbes to depolymerise in the subsoil (Fig. A1), as well as the lower microbial biomass in these layers which strongly reduces the Michaelis-Menten term for the depolymerisation rates (Fig. 1). Additionally, the processes of adsorption and desorption have lower temperature sensitivities than microbial processes in JSM so that warming affects the topsoil layers more strongly than the deeper layers where more SOC is mineral-associated. The soil warming effect is strongest at the beginning of the simulation period, but reduces as the model returns to a new steady state after roughly 100 simulation years.

**Table 2: Model experiments and settings with simulated changes (%) in SOC stocks at three different depth intervals**

| Experiment | ST | SM | $Q_{10,Km,R}$ | $Q_{10,Km,P}$ | Δ SOC (%) 0 - 50 cm | Δ SOC (%) 0 - 6 cm | Δ SOC (%) 36 - 50 cm |
|---|---|---|---|---|---|---|---|
| 1. Ambient model run | + 0.0 K | SM * 1.0 | 1 | 1 | 0 | 0 | 0 |
| 2. Warming experiments | + 4.5 K | SM * 1.0 | 1 | 1 | -5.1 | -6.2 | -3.9 |
| | + 4.5 K | SM * 1.0 | 0.7 | 0.7 | -6.8 | -7.3 | -5.6 |
| | + 4.5 K | SM * 1.0 | 1.3 | 1.3 | -4.0 | -5.5 | -2.8 |
| | + 4.5 K | SM * 1.0 | 0.7 | 1.3 | -5.2 | -6.4 | -3.9 |
| 3. Drought experiments | + 4.5 K | SM * 0.9 | 1 | 1 | -3.0 | -4.4 | -1.9 |
| | + 4.5 K | SM * 0.8 | 1 | 1 | +1.0 | -1.1 | +1.6 |
| | + 4.5 K | SM * 0.7 | 1 | 1 | +6.8 | +3.8 | +7.1 |
| 4. Combined experiments | + 4.5 K | SM * 0.9 | 0.7 | 1.3 | -3.1 | -4.6 | -2.0 |
| | + 4.5 K | SM * 0.8 | 0.7 | 1.3 | +0.5 | -1.5 | +1.2 |
| | + 4.5 K | SM * 0.7 | 0.7 | 1.3 | +5.8 | +3.1 | +6.1 |

### 3.1.2 Temperature sensitivity of half-saturation constants

To study the effects of temperature on SOC decomposition through the half-saturation constant for depolymerisation ($Q_{10,Km,X}$), the model was run using three different combinations of $Q_{10,Km,X}$ values for the depolymerisation of the polymeric litter and microbial residues pools (Table 2): Both $Q_{10,Km,X}$ are 0.7; both $Q_{10,Km,X}$ are 1.3; or $Q_{10,Km,R}$ is 0.7 and $Q_{10,Km,P}$ is 1.3. Using a $Q_{10}$ value for half-saturation constants $K_{m,P}$ and $K_{m,R}$ of 0.7 (reflecting the temperature sensitivity of the depolymerisation of microbial residues) substantially accelerates SOC losses in response to warming. SOC losses until 50 cm depth reach 6.8% (Fig. 3, pink points). The topsoil loses more SOC than the subsoil (-7.3 % and -5.6 %, respectively), but when comparing with the run that uses $Q_{10,Km,X}$ values of 1 the relative difference in the subsoil is larger than in the topsoil. This indicates that in the subsoil, where microbial biomass is lower, the relative importance of $Q_{10,Km,X}$ is larger than in the topsoil. Contrastingly, using a $Q_{10}$ value for half-saturation constants $K_{m,P}$ and $K_{m,R}$ of 1.3 (reflecting the temperature sensitivity of the depolymerisation of the polymeric litter pool) counteracts the warming effect and reduces SOC losses from the soil. The result is still a net loss: SOC stocks in the top 50 cm deplete by 4%, with higher SOC losses from topsoil compared to the subsoil (- 5.5 and - 2.8%, respectively, Fig. 3, orange points). Similar to the model run where $Q_{10,Km,X}$ is 0.7, the temperature sensitivity of $K_m$ in the model run where both $Q_{10,Km,P}$ and $K_{m,R}$ are 1.3 has a relatively larger impact in the subsoil than the topsoil.

Using individual $Q_{10}$ values for the half-saturation constant for the depolymerisation of microbial residues ($K_{m,R}$) of 0.7 and of 1.3 for the polymeric litter pool ($K_{m,P}$) results in SOC losses from the top 50 cm (- 5.2%, Fig.3, yellow). In comparison to the run where $Q_{10,Km,X} = 1$ this result is very similar (5.1% loss from warming alone), indicating that the opposing temperature sensitivities of depolymerisation of microbial residues and polymeric litter pool cancel each other out. The topsoil loses more SOC (- 6.4%) than the subsoil (- 3.9%), but in comparison to the run where $Q_{10,Km,X} = 1$ the losses from the topsoil layer are slightly higher (0.2% additional SOC loss when $Q_{10,Km,X} = 1$) than in the subsoil (no difference).

### 3.2 Drought effects on SOC decomposition

Inducing drought strongly dampens the warming effect on SOC (Fig. 4). For the sake of comparison, Fig. 4 also includes the purple line from Fig. 3 (no drought, SM = SM * 1). Depending on the drought intensity, the top 50 cm lose less SOC, or even act as a sink and start accumulating SOC over the course of the simulation period. A 10% reduction in SM results in a SOC loss of 3%, whereas at 80% and 70% SM, the soil column accumulates 1.0% or 6.8% SOC, respectively (Table 2). Stronger drought intensity led to a larger difference in modelled SOC stocks at 0-50 cm, from -5.1% at the original SM, to + 6.8% at 70% SM, a difference of 11.9 percentage points. This increased drought response on SOC decomposition is a direct result of the increase in the value of $f_{Km,X}(T_{soil}, \theta)$ with decreasing SM ($\theta$, Eq. (2)). Again, the topsoil and subsoil layers show a different response: In the topsoil, there is high microbial biomass and therefore, the effect of the drought on microbial depolymerisation is not as strong as in the subsoil. Additionally, drought decreases the amount of MAOC in the subsoil, while POC accumulates (Fig. A1). As a result, the ratio of POC:MAOC increases, especially in the topsoil (Fig. A2). In 0 - 6 cm, SOC losses are 4.4% at 90% SM, 1.1% at 80% SM, and a 3.8% SOC accumulation at 70% SM. In the subsoil layer between 36 - 50 cm, there is a loss of 1.9% SOC at 90% SM, and 1.6% and 7.1% SOC accumulation at 80% and 70% SM, respectively. At 80% SM the drought has an opposite effect in the topsoil (a net source) than in the subsoil (a net sink). The whole column response, however, is a net sink, highlighting the strong contribution of the subsoil to the overall response.

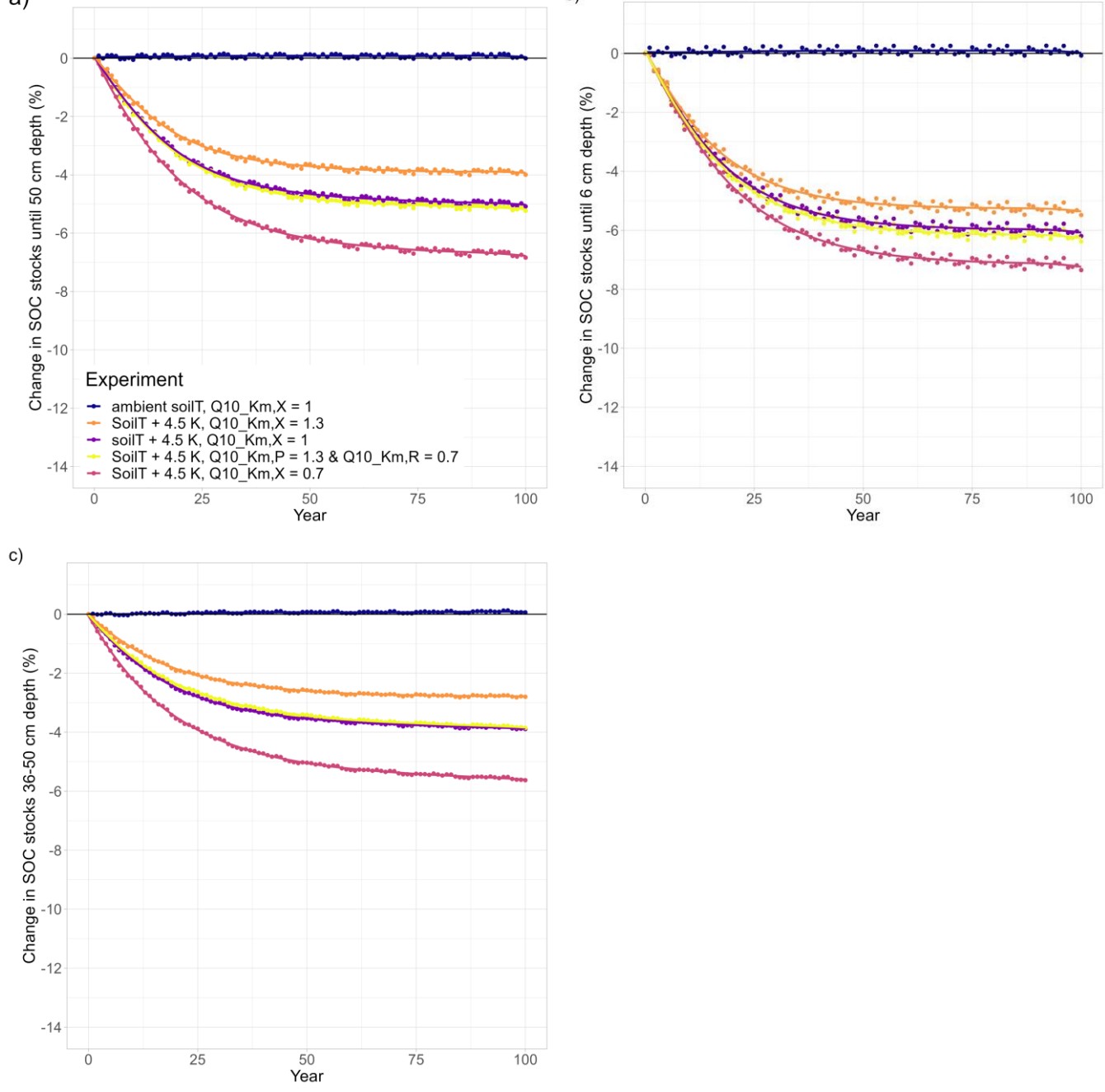

**Figure 3: Temperature effects on long-term changes in modelled SOC stocks (% SOC lost since simulation year 0) in a) a whole soil column (0 - 50 cm), b) the topsoil layer (0 - 6 cm) and c) a subsoil layer (36 - 50 cm). Each subplot shows an ambient model run (dark blue), and the results of four warming experiments (soilT + 4.5 K) with different temperature sensitivities ($Q_{10}$ values) of the half-saturation constants ($Q_{10,Km,X}$). In all runs, ambient SM (SM * 1.0) was used.**

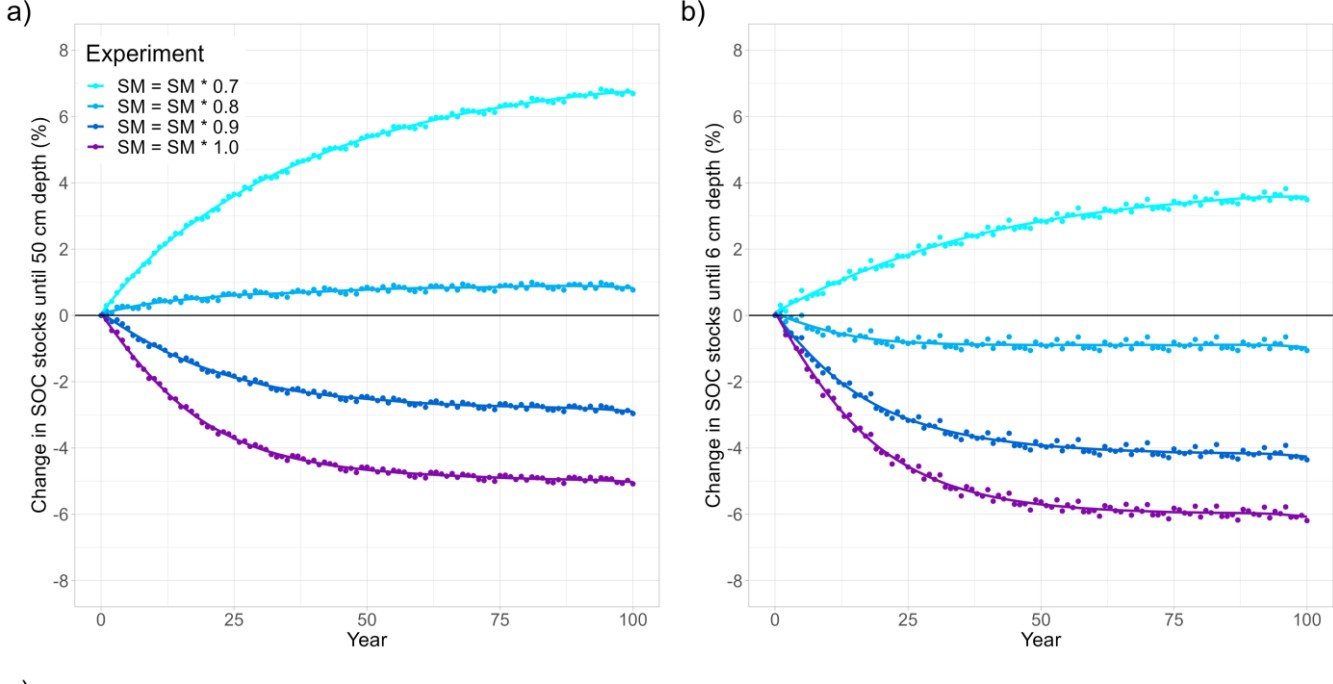

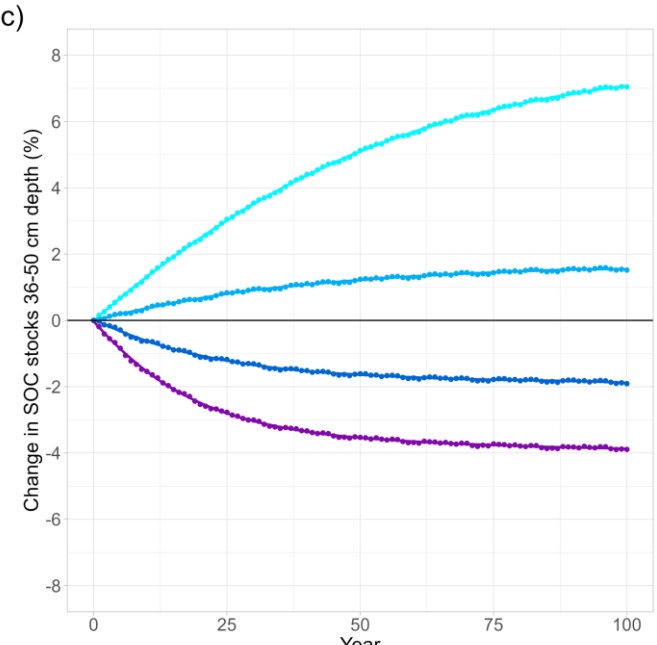

**Figure 4: Soil moisture and temperature effects on long-term changes in modelled SOC stocks (% SOC lost since simulation year 0) in a) a whole soil column (0 - 50 cm), b) the topsoil layer (0 - 6 cm) and c) a subsoil layer (36 - 50 cm). In all model runs, the soil was warmed by 4.5 K. Soil moisture is reduced in 10% steps from ambient (SM * 1.0, purple) to 70% (SM * 0.7, light blue). In all runs, $Q_{10,Km,X}$ = 1 (not temperature sensitive). The purple line (SM = SM * 1.0) is included from Fig. 3 for visual comparison.**

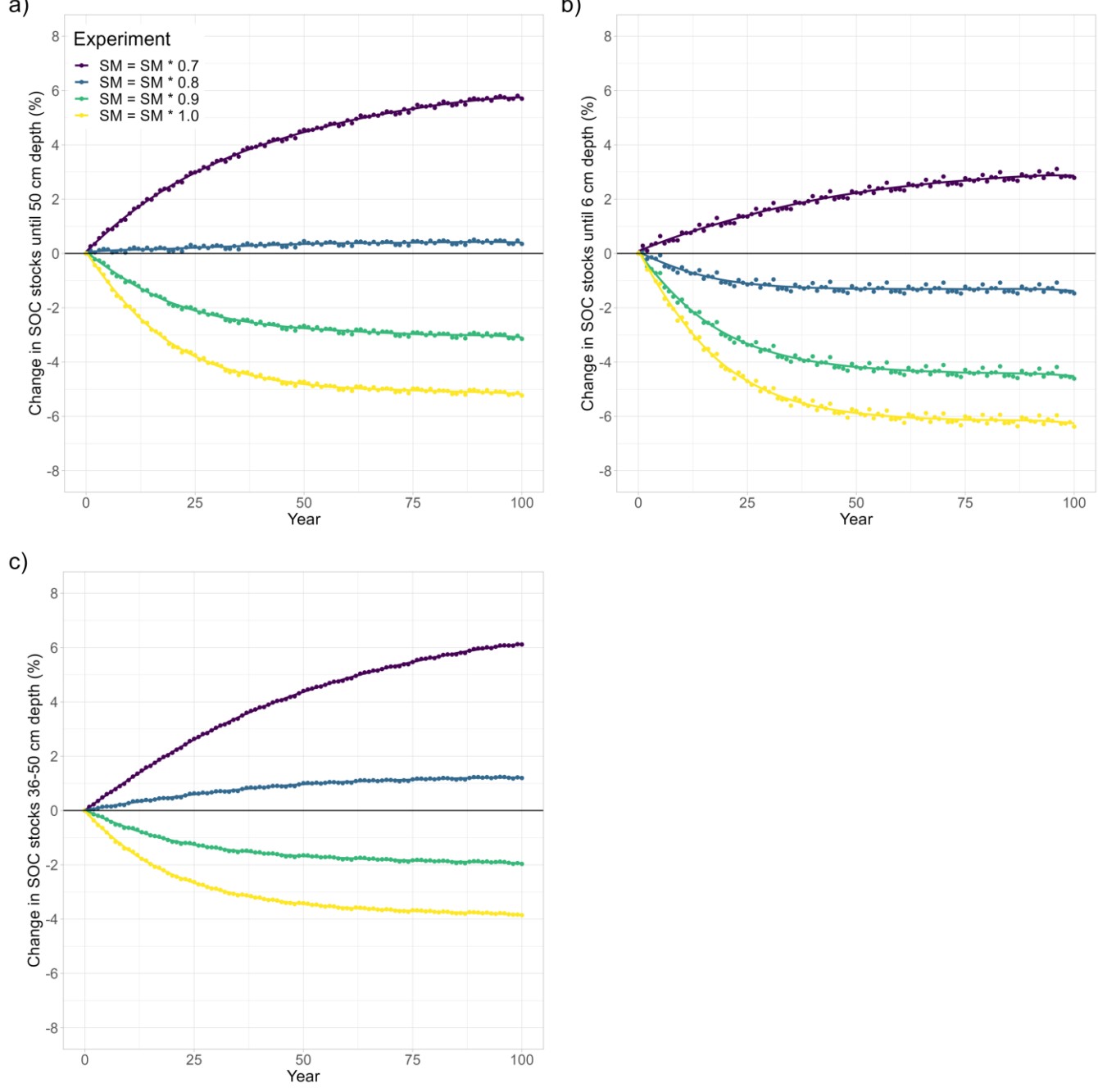

**Figure 5: Temperature and soil moisture effects on long-term changes in modelled SOC stocks (% SOC lost since simulation year 0) in a) a whole soil column (0 - 50 cm), b) the topsoil layer (0 - 6 cm) and c) a subsoil layer (36 - 50 cm). Similar to Fig. 4, the soil**
**was warmed by 4.5 K in all model runs and soil moisture was reduced in 10% steps. Contrasting to Fig. 4 (no temperature sensitivity of the half-saturation constants), $Q_{10,Km,P}$ was 1.3 for depolymerisation of litter and $Q_{10,Km,R}$ was 0.7 for depolymerisation of microbial residues in these model runs. For visual comparison, the yellow line (SM = SM * 1.0) is included from Fig. 3.**

### 3.3 Combined effects of drought and temperature sensitivity of half-saturation constants on SOC decomposition

To investigate the potentially counteracting responses of temperature sensitivity of $K_m$ and droughts, we also run the model for the three different drought intensities in conjunction with a $Q_{10,Km,P}$ value of 1.3 for the polymeric litter pool and a $Q_{10,Km,R}$ value of 0.7 for the microbial residues pool. At ambient SM conditions, the temperature sensitivity of both $K_{m,X}$ values only marginally amplified the warming effects (in Figs. 3 and 5, the yellow points are identical). When SM is reduced, however, this slows down the decomposition rates (Fig. 5, Table 2): At a 10% reduction in available SM, less SOC is lost from the top 50 cm (-3.1%) than when SM is kept at ambient levels (-5.2%). At 80 and 70% SM the soil starts accumulating SOC (+0.5% and 5.8%, respectively). Generally, modelled SOC stocks for this combination of temperature sensitivities of $K_m$ and different drought intensities closely resemble the simulated drought response (Fig. 4), with the temperature sensitivity of $K_m$ counteracting the drought effects: when $K_m$ is temperature sensitive, SOC losses are always higher, and SOC accumulations are always smaller than when $K_m$ is not temperature sensitive (Table 2). Similar to the drought experiments (Section 3.2), the differences in modelled SOC stocks at 0-50 cm increase with stronger drought intensity. Interestingly, the temperature sensitivity effect through $Q_{10,Km,P}$ and $Q_{10,Km,R}$ also increases with stronger drought intensity: For example, at ambient SM, SOC stocks decreased by 5.1% when both $Q_{10,Km,X}$ are 1, and decreased by 5.2% when $Q_{10,Km,R} = 0.7$ and $Q_{10,Km,P} = 1.3$, a difference of only 0.1 percentage points (Table 2). But at 70% SM, SOC stocks increase to 6.8% when both $Q_{10,Km,X}$ are 1, and increase to 5.8% when $Q_{10,Km,R} = 0.7$ and $Q_{10,Km,P} = 1.3$, a difference of 1.0 percentage points. The same trend is also visible for both the topsoil (relative difference from 0.2 percentage point at SM = 1.0 to 0.7 percentage point difference at SM = 0.7) and the subsoil (no difference at SM = 1.0 to 1.0 percentage point difference at SM = 0.7). At the same time, the ratio POC:MAOC did not change much compared to the model run where $K_{m,X}$ was not temperature sensitive (Fig. A2, yellow and pink). This indicates that rather than causing a shift in the litter and microbial residues C pools, microbial limitation is strong under dry conditions (low $C_B$, Fig. 1), which in turn increases the importance of $Q_{10,Km,R}$ and $Q_{10,Km,P}$ for the overall SOC decomposition rates. Contrasting to the results from the isolated warming and drought experiments, the differences in SOC stock changes between the topsoil and subsoil are not very large: From a -6.4% SOC loss to +3.1% SOC accumulation, which is a 9.5 percent point change in the topsoil, to -3.9 to +6.1, in the subsoil (Table 2). These results indicate that the combined sensitivity of SOC stocks to moisture and temperature in topsoil and subsoil is similar due to the counteracting effects of temperature and soil moisture on $K_m$ (Eq. (4): higher temperatures promote SOC decomposition rates due to the stronger influence of depolymerisation of microbial residues (which has a $Q_{10,Km,R}$ value of 0.7), whereas drought decreases SOC decomposition rates.

## 4 Discussion

### 4.1 Warming effects on SOC decomposition

We find that warming the soil by 4.5 K accelerates SOC losses, and these losses are proportionally higher in the topsoil than in the subsoil. This is expected, as higher soil temperature increases maximum depolymerisation rates and microbial growth rates through $Q_{10,Vmax.X}$ (Eq. (2)). Our findings are also consistent with other modelling studies that investigate isolated soil warming effects (Pallandt et al., 2022; Sulman et al., 2018; Todd-Brown et al., 2014; Wieder et al., 2018), as well as with results from a recent large meta-analysis of SOC profiles (Wang et al., 2022a), which reports higher losses of SOC stock and SOC content from topsoil (0-30 cm) than subsoils (0.3 - 1m). In our study, during the warming experiments, the topsoil almost always loses more SOC than the subsoil, except when $Q_{10,Km,P}$ and $Q_{10,Km,R}$ are both set to 0.7 and SOC losses are accelerated in the subsoil (-5.6% loss). Two depth-dependent model processes play an important role in these top- and subsoil differences.

Firstly, microbial biomass ($C_b$) decreases with depth, and as microbial biomass declines, the Michaelis-Menten term for depolymerisation decreases (Fig. 1, Eq. (1)), thereby limiting the depolymerisation rates at lower depths. Secondly, SOC is protected from microbial decomposition by sorption to mineral surfaces, and mineral-associated organic carbon (MAOC, consisting of adsorbed DOC and microbial residues) strongly increases with soil depth (Fig. A1). In JSM, the $Q_{10}$ values of the mineral-associated C pools are 1.08 for adsorption and 1.34 for desorption, which are much lower than the $Q_{10}$ values of the particulate organic carbon (POC) pools: The $Q_{10}$ value for microbial depolymerisation of polymeric litter and microbial residues is 2.16, and 1.98 for microbial C uptake (Table 1; Allison et al., 2010; Wang et al., 2012, 2013). These literature-based values reflect the current understanding that MAOM is less temperature sensitive than POC (Bradford et al., 2016; Tang and Riley, 2015). So, as the ratio of MAOC to POC strongly increases in the subsoil, this leads to an overall lower apparent temperature sensitivity of SOC pools in the subsoil. Total SOC losses consist of DOC, POC, and MAOC from the 36 - 50 cm subsoil layer, so the overall SOC losses may be relatively small as the majority of SOC in this layer consists of protected MAOC - which decreases its overall temperature sensitivity.

Whether or not the apparent temperature sensitivity of SOC declines with depth, as we observe in this study, is still a topic of debate. According to kinetic theory (Bosatta and Ågren, 1999) subsoils may have lower apparent $Q_{10}$ values when they contain less complex, necromass derived substrates (Davidson and Janssens, 2006; Hicks Pries et al., 2023). Contrastingly, the same argument is used to explain higher temperature sensitivities in subsoils when they may contain molecules with higher activation energies (e.g. Li et al., 2020). These observed higher temperature sensitivities could be the result of deriving the apparent $Q_{10}$ values from bulk soil samples containing both POC and MAOC, whereas several other studies demonstrated that this trend can be counteracted by the strong mineral protection of SOC in subsoils (Gentsch et al., 2018; Gillabel et al., 2010; Qin et al., 2019) and that reported high apparent $Q_{10}$ values originate from the decomposition of POC (Soong et al., 2021). In a recent review, Hicks Pries et al. (2023) conclude that the temperature response of deep soils is likely to be context dependent, and that subsoils with high POC content, or with low reactive mineral content are likely to be more susceptible to warming than soils with limited POC or with highly reactive mineral surfaces which protect SOC from microbial decomposition. In our model experiments, MAOC strongly increased with soil depth, which resulted in smaller total SOC losses from the subsoil than the topsoil layer in response to warming.

We observe stronger model responses to different $Q_{10,Km,X}$ values in subsoils than in topsoils, firstly, because microbial limitation is stronger in subsoils than in topsoils. At low microbial biomass ($C_B$), the value of $K_{m,X}$ becomes increasingly important (Fig. 1, Eq. (1)). At the same time, depolymerisation rates only affect the POC pools ($C_P$ and $C_R$) and not the MAOC pools (adsorbed DOC and adsorbed microbial residues). Since the ratio of POC:MAOC is low in subsoils (Fig. A2), the total SOC losses from subsoils are lower than from the topsoil, despite the higher sensitivity to different $Q_{10,Km,X}$ values. So, when $Q_{10,Km,X} < 1$, SOC losses can be further accelerated, especially in the deep soil. In our study, this lower $Q_{10}$ was associated with the breakdown of proteins from the microbial residues pool. The contribution of microbial residues in the deep soil to total SOC is highly significant and can be up to 54% in grasslands (Wang et al., 2021). If free POC in deep soils is indeed more sensitive to warming as a result of low microbial biomass, our model results support the finding that deep soils rich in microbial residues are more temperature sensitive than those that contain less microbially-derived POC contents, due to the lower $Q_{10,Km}$ value of the breakdown of polypeptides. However, compared to plant-derived POC, microbial residues have a high mineral sorption potential (Buckeridge et al., 2022; Liu et al., preprint) and could therefore be more protected from decomposition.

## 4.2 Drought effects on SOC decomposition

Our results show that soil drying can alleviate the losses of SOC from soil warming. In our model, this is the result of the soil moisture sensitivity of the half-saturation constants for microbial depolymerisation ($K_{m,R}$ and $K_{m,P}$, Eq. (4)): Lower soil moisture reduces the Michaelis-Menten term for depolymerisation (Fig. 1), which lowers the SOC decomposition rates. Microbial C uptake for growth is also sensitive to changes in soil moisture through changes in the air-filled pore space (Eqs. 5 and 6), but this would result in faster SOC decomposition rates as microbial growth is less affected by oxygen limitation, which was not the case for any of the modelled drought experiments. Generally, SOC decomposition peaks at intermediate soil moisture, but most soils are below these optimal soil moisture levels and as a result, drying leads to reduced decomposition rates due to stronger microbial limitation, whereas wetting the soil leads to an acceleration of the decomposition rates until oxygen limitation limits SOC decomposition rates (Davidson et al., 2012; Moyano et al., 2018; Pallandt et al., 2022; Skopp et al., 1990; Yan et al., 2018). In our model framework, substrate and oxygen limitation is split between two processes: we simulate moisture-driven diffusion limitation on the microbial depolymerisation rates (reverse MM-kinetics, Eq. (4)), and oxygen and DOC availability affect microbial growth (forward MM-kinetics, Eq. (5)). We found that soil drying consistently reduced modelled SOC losses compared to SOC losses due to soil warming alone, indicating that microbial limitation of depolymerisation is more important than oxygen limitation on microbial growth in our study. Additional support for strong microbial limitation on SOC decomposition comes from our observation that particulate organic C (POC) accumulates in both the topsoil and subsoil layers in response to the most intense drought scenario (SM= SM * 0.7, Fig. A1). If microbes were not limited by the drought, they would degrade POC quickly in response to warming.

Our finding that microbial SOC decomposition consistently declines in response to drought is in agreement with other studies that explore drought effects on SOC decomposition using microbially explicit models (Liang et al., 2021; Wang et al., 2020; Zhang et al., 2022). In the topsoil, we find that the impact of each 10% reduction in SM has a relatively small impact on alleviating SOC losses through warming, compared to the subsoil (Fig. 4). These observed differences in the drought response between top- and subsoil can mainly be explained by the vertical differences in microbial biomass concentration ($C_B$), which is higher in the topsoil than the subsoil. Therefore, at low $C_B$, the relative impact of drought on the MM-term for depolymerisation is larger in the subsoil than the topsoil, making the modelled subsoil SOC stocks more sensitive to drought. For example, at 80% SM, modelled SOC stocks in the topsoil reduce in response to soil warming (from -6.2% to -1.1% as SM reduces to 80%, a net difference of 5.1 percent points), whereas subsoil SOC stocks decrease at ambient SM but increase at 80% SM (from -3.9% to +1.6%, a difference of 5.5 percent points). As discussed in section 4.1, the relatively higher sensitivity of the subsoil to not only warming but also to drought, is also related to the strong increase in MAOC with depth and its lower temperature sensitivity compared to that of the POC pools. In order to focus completely on drought effects on microbial SOC decomposition, adsorption and desorption rates were not sensitive to changes in soil moisture during our experiments. Drought favours the stabilisation of SOC on mineral surfaces (Blankinship and Schimel, 2018) , thereby protecting it from microbial depolymerisation. Therefore, if we would consider the moisture sensitivity of adsorption and desorption rates in our model, we expect a further decrease in the SOC decomposition rates in response to drought. The formulation of moisture sensitivity of adsorption and desorption, however, is not well established to our knowledge.

Overall, our model results indicate a potential for net SOC accumulation in 0 - 50 cm depth when SM is reduced to 80% or 70% of its original values, and that a large part of the whole soil column response is driven by the subsoil. While data-driven deep soil drying studies are rare, our simulation results are supported by a recent study (Brunn et al., 2023), where total annual precipitation throughfall was reduced by 70% for 5 consecutive years and both SOC stocks and SOC stability increased between 0 - 30 cm. They found that the majority of the SOC stock increase occurred in the top 5 cm as a result of higher root

exudates, but we do not consider this in our experiment. We found that the largest SOC stock increase occurred in the subsoil, because of the higher sensitivity of subsoil to drought at low microbial biomass concentrations and the strong protection of MAOC from microbial depolymerisation. Our finding that SOC stocks can potentially increase with drought despite the expected losses through warming, is mainly the result of lower microbial depolymerisation (Eq. (4), Fig. 1). Indeed, short term studies indicate that SOC stocks may increase under drought, as a strong reduction in microbial activity may dominate over the effect of reduced litter and root inputs (Brunn et al., 2023; Deng et al., 2021; Moyano et al., 2013). While results from short-term data-driven studies support our modelling results, long-term drought studies generally show a decline in SOC stocks, which can be mainly attributed to the effects of soil warming and decreased litter inputs (Deng et al., 2021; Meier and Leuschner, 2010). An advantage of our stand-alone soil model environment with prescribed litter inputs is that it allows us to individually test soil warming and drying effects on long-term SOC stocks, while eliminating the potentially confounding effects from changes in plant productivity. Recent research has shown that the chances of drought coinciding with high soil temperatures will further increase in the future (García-García et al., 2023). As a result, the counteracting effects of $K_m$ and drought may be at their strongest, and (semiarid) ecosystems dominated by infrequent moisture inputs may show strong sensitivities to soil warming and drought.

**4.3 Combined effects of moisture and temperature sensitivity of half-saturation constants on SOC decomposition**

Our results indicate that overall, the temperature sensitivity of the half-saturation constants $Q_{10,Km,R}$ and $Q_{10,Km,P}$ had a relatively smaller impact on the overall SOC stock changes (Fig. 5c), compared to the large impact of soil warming on SOC stocks through the temperature sensitivity of the maximum depolymerisation rate, $Q_{10,Vmax.X}$ (Fig. 3c). Such a smaller impact of the half-saturation constants for depolymerisation on SOC stocks was theorised by Wang et al. (2012) if $Q_{10}$ values for $K_m$ are close to 1, as is the case in our study. Saifuddin et al. (2021) also reported a small impact of $Q_{10,Km}$ versus $Q_{10,Vmax}$ on simulated SOC stocks, but their model study did not consider multiple soil depths, the effects of drought, and organo-mineral interactions. In this study, we show that soil drying in combination with temperature sensitivity of the half-saturation constants for depolymerisation of polymeric litter and microbial residues, can both increase or decrease SOC stocks, and that the direction and magnitude of the effect on SOC stocks depends on drought intensity. The combined effects of soil drying and temperature sensitivity of the half-saturation constants for depolymerisation on SOC stocks closely resembled that of the drought response, which indicates that microbial limitation on depolymerisation poses a strong control on modelled SOC stocks and that drought can indeed alleviate SOC losses in response to soil warming. While the effect of drought on modelled SOC stocks is strong, the temperature sensitivity of $K_{m,X}$ can counteract these effects: Compared to the model runs without temperature sensitivity of $K_{m,X}$, SOC losses are higher and SOC accumulations are smaller. This indicates that the breakdown of microbial residues, which had a $Q_{10,Km,R}$ value of 0.7, is important for the overall results because a $Q_{10}$ value lower than 1 increases the MM-term for depolymerisation, and accelerates SOC decomposition. Furthermore, this counteracting effect of $Q_{10,Km,X}$ is stronger with increased drought intensity while the ratio POC:MAOC does not change much when compared to the model run where $Q_{10,Km,X}$ is not temperature sensitive. In line with our results from the isolated drought experiments (Section 4.2), this supports the conclusion that microbial limitation increases under drought, so that $Q_{10,Km,X}$ becomes more important for the overall depolymerisation rates.

Unlike the individual warming and drought experiments, we only find small differences in SOC stock changes between the top- and subsoil for the combination of drought and temperature sensitivity of $K_{m,R}$ and $K_{m,P}$. This shows that drought and temperature sensitivity can both play a strong role, and counteract each other so that the overall changes in SOC stocks appear similar. This is an important result, because long-term warming can accelerate soil drying, especially at the soil surface (Berg

and Sheffield, 2018; Fan et al., 2022; García-García et al., 2023). Our results show divergent responses of top and subsoil SOC stocks to concurrent soil warming and drying, in particular at a 20% SM reduction, where modelled SOC stocks in the topsoil increase but decrease in the subsoil. While we only explored the effects of evenly drying out the soil column in this study, the long-term response of SOC stocks to soil moisture changes could be different as top- and subsoils may not dry out evenly (Berg et al., 2017). Using multi-model predictions, Berg et al. (2017) show that surface soil moisture decreases by the end of the century, while subsoils, especially in the northern hemisphere, diverge with either less severe drying or wetter conditions. On top of soil warming, such dynamic vertical changes in soil moisture have a strong potential of further accelerating or slowing down SOC decomposition rates in the deep soil by microbial limitation or oxygen diffusion limitation (Pallandt et al., 2022). We call for modelling studies that address such changes simultaneously by running NGSMs with future climate forcing datasets.

**4.4 Microbial response to substrate changes in the POC/MAOC framework**

The duration of our experiments is 100 simulation years, but the values of $Q_{10,Vmax}$ and $Q_{10,Km,X}$ may not stay constant over time, as the environment changes and microbial communities adapt. However, in light of our long-term warming experiments we feel confident with the choice of $Q_{10,Km}$ values, as they were measured in microorganisms that showed no sensitivity to a 6 K increase in average temperature - but did show a strong response to changes in substrate types (Allison et al., 2018a). In our model experiments, microbes have access to both litter inputs and microbial residues to depolymerise, which have counteracting $Q_{10,Km}$ values and therefore the possibility to simultaneously accelerate and slow down microbial SOC decomposition rates. Therefore, it would be useful to consider soils that are high in POC versus soils that are high in MAOC: Soils with high MAOC contents and low POC inputs can have lower apparent $Q_{10}$ values because $Q_{10,sorption}$ is much lower than the $Q_{10}$ values of unprotected organic carbon (Table 1; Wang et al., 2012, 2013). Such soils would have necromass rather than fresh litter inputs as the dominant C substrate for microbes. New datasets such as global maps of necromass C contributions to total SOC stocks (e.g. Liu et al., preprint) can inform modellers on substrate type or SOC stabilisation mechanisms, and thereby help identify the climate sensitivities of SOC stocks in different regions of the world. At the moment, though, there are no clear answers as to which values we should use for $Q_{10,Km,X}$ because SOC consists of many different molecules, which all have their own specific temperature sensitivities (Allison et al., 2018b). One possibility to investigate the potential climate-substrate feedbacks with a model like JSM, would be a further partitioning of the litter pools into functional groups related to their main degrading enzymes. Our current study, which explores different values for $Q_{10,Km,X}$ already provides valuable insights into what might be possible. For example, soils with high POC contents, i.e., with a developed organic layer as a result of high litter inputs, low SOC losses and low bioturbation, are likely to have $Q_{10,Km,X}$ values > 1, which has the potential to counteract soil warming effects through $Q_{10,Vmax}$, especially in deeper layers where microbial biomass is low and the temperature sensitivity of the half-saturation constant will have a stronger impact. In combination with soil drought, this would further enhance microbial limitation for depolymerisation and could dampen SOC losses in such organic soil layers over time - if litter inputs stay constant over time. Peat soils could be an exception, as they usually have high volumetric water contents and reduced soil moisture can lift oxygen limitation, thereby increasing SOC decomposition rates. It can be expected though, that long-term soil drying reduces root and leaf litter inputs as plant productivity decreases (Deng et al., 2021). Plants and microbes could also actively compete for nutrients in such a coupled model environment, which, apart from the climatic sensitivities investigated in this study, will introduce additional non-linear feedbacks on future SOC dynamics through changes in e.g. litter inputs and microbial Carbon Use Efficiency (Braghiere et al., 2022; Thurner et al., 2024). Therefore, we recommend future research focuses on further studying climate-substrate interactions within a fully

coupled soil-plant model, such as the coupling between land surface model QUINCY (Thum et al., 2019) with JSM, which is nearing completion.

## 5 Conclusions

With our JSM model experiments, we show that both soil drying and warming pose strong controls on SOC decomposition. The vertically explicit model structure allows us to demonstrate that subsoil SOC stocks respond differently to warming and drought through a combination of processes. First of all, we show that SOC association to mineral surfaces plays an important role in reducing the overall sensitivity of SOC stocks to microbial decomposition: MAOC strongly increases with soil depth and has a low apparent temperature sensitivity, which results in smaller total SOC losses from the subsoil than the topsoil. At the same time, our model results indicate that unprotected subsoil SOC is more sensitive to soil warming and drought. Secondly, we show that drought can alleviate the effects of soil warming through microbial limitation on depolymerisation rates. As drought gets stronger, microbially mediated depolymerisation rates become severely limited so that less SOC is lost from the soil. In the model experiments with constant litter inputs in this study this can even lead to SOC accumulation over time, despite soil warming. Thirdly, we show that considering the temperature sensitivities of the half-saturation constants for different C substrates (litter and microbial residues) is important, as they can both slow down and accelerate microbial SOC decomposition rates. Our results highlight the importance of representing SOC decomposition processes in a vertically resolved model, which includes carbon stabilisation on mineral surfaces. We recommend that future model development focuses on further identifying the (un)importance of temperature sensitivities of $V_{max}$ and $K_{m,X}$ for different C substrates and moisture sensitivities of all microbial-mineral interactions in the new class of soil organic carbon models.

**Appendix A**

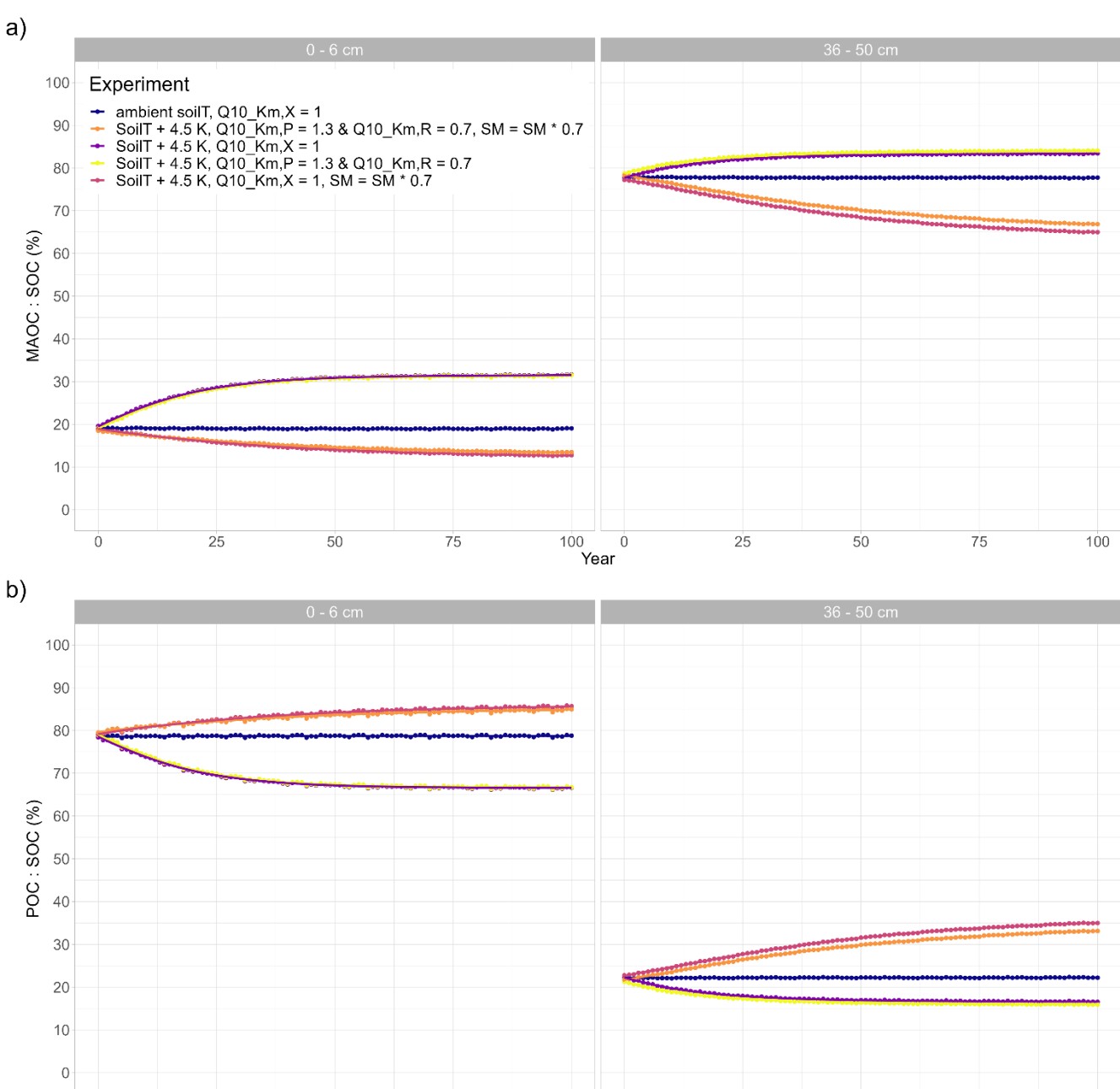

Figure A1: a) Ratio of mineral associated organic carbon (MAOC) to SOC (%) and b) particular organic carbon (POC) to SOC (%) for different model runs at two different soil depths: Topsoil (0 - 6 cm) and subsoil (36-50 cm). If not indicated otherwise, SM = SM * 1.0 in the experiment.

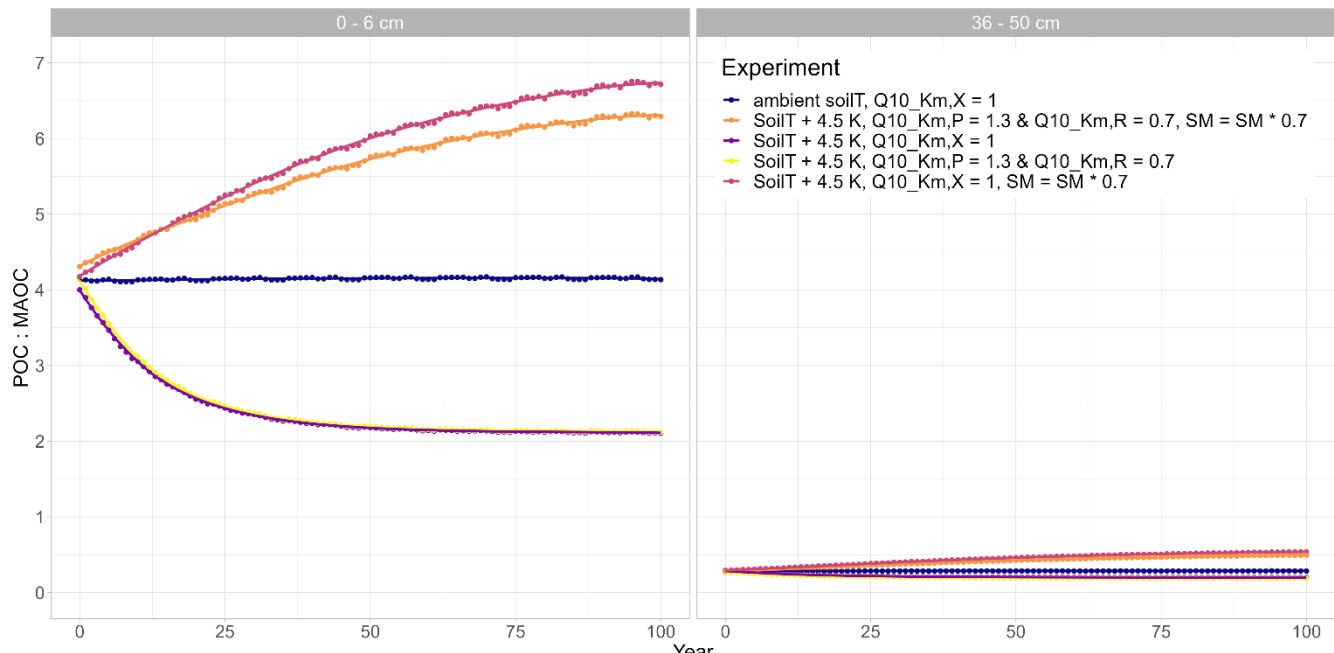

**Figure A2: Ratio of particular organic carbon (POC) to mineral-associated carbon (MAOC) for different model runs at two different soil depths: Topsoil (0 - 6 cm) and subsoil (36-50 cm). If not indicated otherwise, SM = SM * 1.0 in the experiment.**

580

**Code availability**

The Jena Soil Model (JSM) - release01 is fully described and published under https://doi.org/10.5194/gmd-13-783-2020. The JSM source code is available online (https://git.bgc-jena.mpg.de/quincy/quincy-model-releases, branch "jsm/release01"), but access is restricted to registered users. Readers interested in running the model should request a username and password from S. Zaehle or via the git repository. JSM is developed using the framework of the QUINCY model. The QUINCY model is free software: It can be distributed and/or modified under the terms of the GNU GPL version 3 (https://www.gnu.org/licenses/gpl-3.0.en.html, last access 10 January 2024). The use of the QUINCY model relies on the application of software developed by the MPI for Meteorology, which is subject to the MPI-M ICON software licence (see ICON section: "By using ICON, the user accepts the individual licence (https://code.mpimet.mpg.de/attachments/download/20888/MPI-M-ICONLizenzvertragV2.6.pdf, last access 30 August 2024). Where software is supplied by third parties such as the MPI for Meteorology, it is indicated in the header of the file. Model users are strongly encouraged to follow the fair-use policy stated at https://www.bgc-jena.mpg.de/en/bsi/projects/quincy/software (last access: 30 August 2024).

**Author contribution**

MP wrote the original draft for the manuscript with contributions from all co-authors. BA and LY wrote the original JSM model code (version 1.0, see code availability), MP and BA wrote minor code modifications to run the model experiments as described in Section 2, and MP performed the model simulations and created all visualisations. This study was conceptualised by MP, BA, MS, and HL. Formal analysis was carried out by MP and BA. Funding acquisition: BA, HL and MR. Supervision of MP by BA, HL, MS, and MR.

**Acknowledgements**

MP is grateful for discussions with Prof. S. Zaehle in the early phases of this study as well as technical support from J. Nabel and J. Engel from the QUINCY modelling team at the Max Planck Institute for Biogeochemistry in Jena. MP received funding support for this work from the Norwegian Research Council through grant no. RCN 255 061 (MOisture dynamics and CArbon sequestration in BOReal Soils) and the Max Planck Institute for Biogeochemistry in Jena. During the revisions of the manuscript, MP was supported by Stockholm University and the European commission through the project 'Holistic management practices, modelling and monitoring for European forest soils' - HoliSoils (EU Horizon 2020 Grant Agreement No. 101000289).

**Competing interests**

The authors declare that they have no conflict of interest.

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
