# Peer review of "Modelling climate-substrate interactions on soil organic matter decomposition with the Jena Soil Model"

_EGUsphere, 2024_

## Author Comment (AC1)

**Response to Anonymous Referee #1**

The study by Pallant et al. aims to investigate the interactions between microbial depolymerization and climate change, specifically the influence of temperature and moisture on SOC decomposition and the differential response of soil layers to warming and drought, using the JSM model. Despite the study's promising title and significant potential contributions to future SOC research, it falls short in several key areas, preventing it from being a comprehensive study. In its present state, it seems incomplete, suitable at best for a sensitivity analysis within a broader, more thorough investigation. I recommend withholding publication until substantial improvements are made.

*We thank the reviewer for taking the time to read and comment on our manuscript. Below we provide detailed comments to the reviewer's questions and suggestions, and describe the planned improvements for the updated manuscript. We apologise for the apparent mismatch between the title and the content, and will adjust the title for the updated manuscript. Additionally, the updated manuscript will have clearer descriptions of the study's objectives, novelty and scope, in line with our response to the comments and suggestions of the other anonymous reviewer.*

**General comments:**

The title's implication of a drought-focused study misaligns with the content, which predominantly consists of sensitivity tests devoid of any empirical assessment. The absence of observational data reduces the study to hypothetical model outputs, with an overemphasis on parameter sensitivity to temperature and moisture.

*We regret the apparent mismatch between the title and the content, and will adjust the title to "Modelling climate-substrate interactions on soil organic matter decomposition with the Jena Soil Model" so that it better reflects the paper's scope.*

*This is indeed a modelling study with a strong focus on the sensitivities of depolymerisation rates to changes in temperature and soil moisture changes. These are processes that have partially been overlooked in dynamic modelling studies before, in particular the soil moisture impact. In this work, we aim to study the effect of these temperature and moisture sensitivities parameterized based on data from a lab study by Allison et al. (2018). While JSM is a relatively new model, the processes and parameters for JSM and its predecessor COMISSION have been successfully tested against observations for various applications by Yu et al. (2020, 2023), Ahrens et al. (2015, 2020), and Fleischer et al. (in prep, but see [https://meetingorganizer.copernicus.org/EGU22/EGU22-11276.html](https://meetingorganizer.copernicus.org/EGU22/EGU22-11276.html)). Making frequent reference to these existing studies, we are not conducting additional comparisons between model outputs and observational data in this study. It also would be difficult to do here, since e.g. available respiration data would only cover the first few years of the simulations for the experiments, and SOC measurements from warming/drought experiments would be highly impacted by changes in plant productivity (litter inputs). A discussion of this study limitation is already part of the original manuscript.*

This study also overlooks other crucial dynamics in SOC, such as diffusion and advection, which could significantly influence SOC distribution within soil columns and thus its response

to soil temperature and moisture changes. The omission of factors like soil texture and vegetation dynamics further limits the study's scope.

*JSM as it was applied in this study already includes the above-mentioned processes, which are described in Yu et al. (2020) and Ahrens et al. (2015, 2020). Diffusion, advection, and soil texture are all considered by the model. Soil texture influences soil moisture and water fluxes but also the sorption of DOM and microbial residues via the maximum sorption capacity, Qmax. For the sake of brevity, these are not described in detail in this manuscript, but we will mention the inclusion of these processes in the methodology section in the updated version of this manuscript. The aim of our study is to investigate in detail the effects of interacting temperature and moisture sensitivities in a dynamic Michaelis-Menten term. To our knowledge, this has not been done before in a comprehensive manner and this warrants an in-depth modelling study. This novel aspect of our study will be better highlighted in the introduction of the updated manuscript.*

*To fit the objectives for this study, we made the conscious choice to not have an active response of the vegetation to moisture and temperature changes and use constant litter forcing files over time, to be able to pinpoint the changes in modelled SOC dynamics in response to the climatic drivers in isolation from those driven by changes in litter inputs.*

The introduction should be revised to avoid methodological details and instead provide a comprehensive overview of the factors controlling SOC dynamics. Additionally, the methodology does not adequately discuss SOC discretization or the interplay between the topsoil and subsoil layers.

*We feel that a comprehensive overview of the factors controlling SOC dynamics falls beyond the scope of the introduction, since we want to focus on the interplay of soil temperature and soil moisture sensitivities in a microbial-mineral model at different soil depths. We will add details on processes that are regulated by temperature and moisture and refer to previously published work with JSM for other controlling factors of SOC decomposition that are beyond the temperature and moisture sensitivity scope of this manuscript. We agree with the reviewer that the methodology section should include a more thorough description of the interplay between top and subsoil layers. This will be changed in the updated version of the manuscript, where also the descriptions of SOC advection and diffusion mentioned in the previous comment can be included.*

I have confined my comments to the methodology section because the paper's foundation needs strengthening before the results and discussion can be meaningful.

**Specific comments:**

L20: Clarify the term "long-term" to provide proper context.

*We change to "We find that soil warming leads to SOC losses after 100 simulation years"*

L20: Define "SOC-specific Q10" to elucidate its relevance to the study.

*We clarify this in the abstract by addressing the Q10Km value for plant litter (1.3) and Q10Km for microbial necromass (0.7) in the abstract. These two Q10s have opposing effects*

*on decomposition and interact with the amount of microbial biomass via the Michaelis-Menten term. This introduces a different sensitivity between topsoil and subsoil as microbial biomass decreases with the amount of root litter inputs and advective and diffusive transport inputs which decrease with depth.*

L21: Distinguish between "reduce" and "accelerated" processes to prevent ambiguity.

*We rephrased to: We find that soil warming increases maximum depolymerization rates, but depending on the share of plant litter and microbial biomass, these increased maximum depolymerization rates can be in full effect when microbial biomass is high, as in the topsoil or microbial necromass makes up the majority of SOC (accelerating Q10KM). On the other hand, SOC through increased maximum depolymerization rates are reduced when microbial biomass is low or plant litter makes up the majority of SOC (decelerating Q10).*

L23: Replace "SOC gain" with "SOC accumulation" to accurately reflect changes in SOC cycling during different environmental conditions.

*We rephrase to SOC accumulation*

L24: When using the term "decomposition rate," additional details should be provided for clarity. It might be preferable to use "SOC loss" for simplicity and better understanding.

*We rephrase to SOC balance instead of SOC loss, since it can be both a SOC gain or loss.*

L25-26: You need to be clearer. Maintain consistency in the comparison of temperature and moisture effects on both topsoil and subsoil.

*We rephrase to "In this study, we show that while absolute changes to the SOC balance in response to warming and drought are highest in the topsoil, SOC in the subsoil is more sensitive to these changes due to the intricate interplay between Km, temperature, soil moisture and mineral-associated SOC."*

L28-29: These assertions require supporting references for validation

*We add the missing references and include Fan et al. (2022), Crowther et al. (2019) in the first part, as well as Davidson (2020) and Kirschbaum (2006) in the second part.*

L34-35: The phrase "most important" is subjective and should be reworded to reflect that soil temperature and moisture are significant controlling factors. This section would benefit from an expanded discussion on the key factors that regulate soil dynamics.

*We agree the wording can be perceived as subjective and rephrase to "Soil temperature and soil moisture are significant controlling factors of microbial decomposition rates, and thereby the carbon turnover rate of soils (Davidson and Janssens, 2006; Moyano et al., 2013; Yan et al., 2018)." Additionally, we put higher emphasis on the importance of SOC stabilisation on mineral surfaces as another important factor. In this section (lines 33 - 37), we emphasise the most important factors that fall within the scope of this study.*

L35: Clarify the term "de(stabilisation)" for better understanding.

*We include "microbial SOC decomposition, and adsorption and desorption of SOC."*

L37-39: Reformulate these lines for enhanced clarity and readability.

*We apologise for unclear wording and rephrase to "Given the importance of SOC stocks and their sensitivities to climate change, a better understanding and representation of these complex interactions in models is extremely important for a better understanding of the carbon-climate feedback"*

L41: Specify the models referred to in this line.

*We rephrase to "Tang and Riley (2019) showed that the microbial depolymerisation rate can be described using reverse Michaelis-Menten (MM) kinetics."*

L43-80: Shift the detailed methodological exposition to the methods section, using the introduction to provide a broader overview.

*This simple description of the reverse Michaelis-Menten kinetics with equations and the conceptual figure is a key component of the introduction. To our knowledge, previous works have primarily focussed on the temperature sensitivities of Vmax (or the apparent sensitivities of (heterotrophic) respiration rates). The novelty of our study is the focus on the temperature and moisture sensitivities on depolymerisation rates through the half-saturation constant (Km), which to our knowledge has not been shown in a modelling study before. As reverse Michaelis-Menten kinetics (compared to forward kinetics) may be less known to the readers of Biogeosciences, we think the equations and conceptual figure help clarify why microbial depolymerisation rates can both increase or decrease in response to soil warming and drought/soil wetting through Km, especially at low microbial biomass concentrations.*

L83: A reference is missing and should be included.

*Many studies have shown this, but we refer to the recent comprehensive review on deep soil carbon by Hicks Pries (2023).*

L88: What about peatlands? Inclusion or exclusion of them would significantly change your findings. It is important to note that the moisture function, decomposition term, and vertical dynamics of SOC distinctly vary in wetland environments.

*Yes, this is definitely true for peatlands, and the traditional empirical model approaches we refer to here also don't describe SOC-moisture-temperature dynamics in these ecosystems very well (or even do consider them at all, see e.g. Chadburn et al. (2022)). Our results are for a mineral soil profile, so we feel explicitly discussing peatlands in the introduction falls outside of the scope. We will mention that our modelling approach covers upland mineral soils in the introduction. Peatlands and their different moisture dynamics, however, are briefly discussed at the end of Section 4.4.*

L95-99: Avoid repeating information from the methods section to streamline the text.

*We will remove and/or move redundant information on JSM.*

L113: Incorporate the previously mentioned introductory information here for coherence.

*As mentioned in the earlier comments above, we will include additional information for JSM on the processes of diffusion, advection, and soil texture, with textual edits and an update of Fig. 2 and its caption (see comment below on L116).*

L115: Briefly explain each parameter, including those in Table 1, and cite the source of the formulations. Additionally, explain the rational for such formulation for DOC?

*This formulation describes the depolymerisation of litter and microbes to DOC. Hence, it is not a formulation for DOC. The formulation is based on the derivations by Tang and Riley that we mentioned in the previous sentence. We apologise for the unclear language and rephrase.*

Eq1: For this and all subsequent equations, it is necessary to specify the units of the calculated flux. Additionally, clarify if any conversions were performed to derive these units from the parameters listed in Table 1.

*We will add the units of all fluxes defined for Eqs. 1 – 7 and apologise for some small typesetting mistakes we found in the superscript of the units in Table 1.*

L116: Although the layered structure of the soil has been mentioned and is depicted in Figure 2, there is a noticeable absence of any depth indicators in the equations or in the estimations provided.

*We will add a clarification in the methods section to explain that all C pools and forcings are functions of time and depth, e.g. Tsoil (t,z), $C_B(t,z)$. In addition, this will be made clear in the updated Fig. 2.*

Figure 2: The figure needs to be revised for clarity, including an explanation of the brown arrows and a clear differentiation between soil pools and depth indicators.

*We will revise Fig. 2: The brown arrows represent the above and belowground inputs. We separate these labels into 'aboveground litter' and 'root litter', and move the label for the belowground arrows (root litter inputs) to the left. Every model pool is present in each depth. We clarify in the caption that this is a vertical model structure and the pools are just zoomed in for one layer as an example.*

Table 1: Complete the table by labelling parameters that lack units as "unitless." Additionally, is there a reference for the parameter R that can be provided?

*R, the universal gas constant, is not a parameter but a well-known constant, as mentioned under Eq. 5, L.139. We report the universal gas constant in $J\ K^{-1}\ mol^{-1}$, adhering to the journal's use of standard SI units. We add 'unitless' to Table 1 where applicable, and also update the heading from "Parameter"to "Parameter/constant" for clarity.*

L145: The nutrient dynamics have a significant influence on uptake and other soil processes. While you state that the model accounts for nutrients, there's a missed opportunity to thoroughly explore and discuss this aspect within the current study.

*We focus on one novel aspect in this study: the interaction of soil moisture and soil temperature sensitivity of a dynamic Michaelis-Menten term. We add a discussion on how*

*microbial carbon use efficiency (CUE) may be temperature sensitive. Carbon use efficiency is one major process that can be influenced by nutrients and temperature. The temperature sensitivity of CUE is, however, not well described empirically and even less so the moisture sensitivity of CUE. Extending this work by including nutrient dynamics and its temperature and moisture sensitivity leads to a path far outside the domain covered by our study's model approach.*

L147: Consideration of root distribution across various soil depths is missing. An analysis of how roots are spread throughout the soil profile could offer valuable insights into soil dynamics.

*We apologise for not clearly communicating that we consider root distribution across various root depths (see revised Fig. 2). The root distribution is the main factor for different microbial biomass levels in different depths (Ahrens et al., 2020).*

L153: Include the sorption/desorption equation to complete the methodological description.

*Sorption and desorption are clearly described in Ahrens et al. (2015, 2020), which we refer to here. While checking this sentence we found a small mistake in the description of the activation energies, which we intend to correct: All Q10 values used in this study are based on literature, and not calibrated.*

L184: What do you mean by first model run? Explain the rationale behind the first model run and the selection of a 100-year period.

*We rephrase this section and start by describing the spin-up and then the 'first' model run, which is now labelled as the ambient model run to be consistent with the rest of the manuscript.*

L185: The decision to focus solely on the 0-50 cm soil layer raises questions. Should processes like advection or diffusion be considered, the entire soil column to its maximum depth would likely experience changes during the spin-up period. Why then, is the analysis limited to this specific depth range?

*The processes of advection of DOC and diffusion to represent bioturbation are considered in JSM. In JSM, there are 15 soil layers, of which 6 layers between 0 and 50 cm depth, and we will make this more clear in the methodology section. These 6 layers experience the strongest influence of transport. Soil layer thickness increases with increasing soil depth, at the following depth intervals: 0 – 6, 6 – 13, 13 – 20, 20 – 26, 26 – 36, and 30 – 50 cm. So we did not focus on only one layer between 0 – 50 cm, but compared the processes in the vertical profile setting with multiple soil layers. By comparing a shallow and a deeper layer, we were able to look at the differences along a gradient of microbial activity – the majority of microbial activity and the resulting heterotrophic respiration flux takes place in the upper soil layers. Additionally, the difference between the shallow and deeper layer illustrate the effect of decreasing root litter inputs with depth.*

*JSM also provides model outputs below 50 cm, which were checked but not shown in the manuscript. These deeper soil layers indeed take many years to reach equilibrium, as the reviewer points out. So another, more practical reason for choosing to focus on the layers*

*until 50 cm depth is that this significantly reduces the necessary model spin-up time and thereby the use of computational resources. Despite these layers not being in steady state after spinup period, we did check the patterns in the layers between 50 and 100 cm depth and found they are similar to what we show in the manuscript. We also found that for the drought and warming experiments, these deeper layers needed more than 100 simulation years to reach a new steady state after disturbance, but they did eventually reach it.*

L186: Detail the approach used to assess steady-state conditions.

*We used a linear regression model to check the steady state assumptions and will mention this in the methods. It is also visually very clear from the model results, e.g. in Figure 3.*

L189: Clarification is needed on the term "all ambient soil temperature." Does this refer to temperatures at every soil depth? Moreover, to isolate the impact of temperature alone, it would be necessary to perform simulations with varying temperature and moisture, and then compare these to simulations with only temperature variation to accurately determine temperature sensitivity.

*We apologise for unclear wording. Temperatures are depth-specific, which will be better clarified in the methods section. These depth-specific temperatures were increased by 4.5 K for each depth. Similarly, the depth-specific moisture values were decreased by 10, 20 or 30% in the different drought experiments. We did consider the isolated effects of soil warming or drought by including model experiments where either temperature or soil moisture was kept at its original, depth-specific input values.*

L194-196: Justify the chosen experimental values.

*The full justification for these values was provided in the original manuscript in lines 166-183, under a separate section of the manuscript titled "2.3 Choice of Q10,Km values for polymeric litter and microbial residues''. In short, the values are based on a lab study on the temperature sensitivities for different enzymes involved in the breakdown of soil carbon substrates (Allison et al., 2018). The Q10,Km value for the depolymerisation of litter was based on measurements for the enzymes β-xylosidase and total oxidase, and the Q10,Km value for the depolymerisation of microbial residues was based on measurements for the enzyme leucine aminopeptidase.*

L201: What was the rational for 10% changes and why only between 0.9-0.7 values?

*We apologise for not clarifying this in the methodology section. Part of the rationale was somewhat hidden in the introduction, and we will restructure the document and state the rationale in L201. We used the following rationale:*

*For projected soil moisture changes there is much uncertainty in model projections: 60 projected global lateral and vertical distributions of future soil moisture are highly diverse both in the predicted spatial and vertical distributions of future soil moisture (Berg et al., 2017). We chose the strengths of the drought intensity (up to 30% reduction in SM) based on the ranges of Berg et al. (2017) Figures 3 and 4, showing the multi-model mean and medians for subsurface and deep surface soil moisture changes by 2100. Given the divergence between models, we chose to simulate drought by simply reducing the depth-*

*specific soil moisture values from the forcing dataset. We chose 10% changes for clarity, to be able to compare the effects of a relatively mild versus increasingly stronger droughts.*

L210: Confirm whether equilibrium was reached after 100 spins and discuss mass balance and steady-state conditions for each run.

*We used a linear regression model to check the steady state assumptions and expand the text describing the steady-state check in the methodology (section 2.4). Mass balance checks are performed in the QUINCY-JSM model code, and for every run we have log files showing that the mass balance is closed. Furthermore, the QUINCY-JSM model is part of the EucFACE-MIP data-model comparison project, and a reference (in revision with Science Advances) to the detailed descriptions of the mass balance checks for C, N and P is:*
***Title: Carbon-phosphorus cycle models over-estimate CO$_2$ enrichment response in a mature Eucalyptus forest***

***Authors:** Mingkai Jiang [1,2*], Belinda E. Medlyn [2], David Wårlind [3], Jürgen Knauer [2,4], Katrin Fleischer [5], Daniel S. Goll [6], Stefan Olin [3], Xiaojuan Yang [7], Lin Yu [5,8], Sönke Zaehle [5], Haicheng Zhang [9], He Lv [1], Kristine Y. Crous [2], Yolima Carrillo [2], Catriona Macdonald [2], Ian Anderson [2], Matthias M. Boer [2], Mark Farrell [10], Andrew Gherlenda [2], Laura Castañeda-Gómez [11], Shun Hasegawa [2,12], Klaus Jarosch [13,14,15], Paul Milham [2], Raúl Ochoa-Hueso [16,17], Varsha Pathare [2,18], Johanna Pihlblad [2,19,20], Juan Piñeiro Nevado [2,21], Jeff Powell [2], Sally A. Power [2], Peter Reich [2,22,23], Markus Riegler [2], David S. Ellsworth [2], Benjamin Smith [2]*

L216: The mention of R packages is less critical than a thorough analysis of the outputs. Consider focusing on the methods to output evaluation here.

*We expand this section with a short description of the smoothed curve shown in Figures 3-5 which visually support the data points plotted for the 100 simulation years. We strongly feel that the mentioning of R packages with citations are important and should remain part of the manuscript, as we are grateful to the many (scientific) open source software developers for their important contributions, and acknowledge this by crediting their work with citations.*

*References:*

*Ahrens, B., Braakhekke, M. C., Guggenberger, G., Schrumpf, M., and Reichstein, M.: Contribution of sorption, DOC transport and microbial interactions to the 14C age of a soil organic carbon profile: Insights from a calibrated process model, Soil Biol. Biochem., 88, 390–402, https://doi.org/10.1016/j.soilbio.2015.06.008, 2015.*

*Ahrens, B., Guggenberger, G., Rethemeyer, J., John, S., Marschner, B., Heinze, S., Angst, G., Mueller, C. W., Kögel-Knabner, I., Leuschner, C., Hertel, D., Bachmann, J., Reichstein, M., and Schrumpf, M.: Combination of energy limitation and sorption capacity explains 14C depth gradients, Soil Biol. Biochem., 148, 107912, https://doi.org/10.1016/j.soilbio.2020.107912, 2020.*

*Allison, S. D., Romero-Olivares, A. L., Lu, Y., Taylor, J. W., and Treseder, K. K.: Temperature sensitivities of extracellular enzyme Vmax and Km across thermal environments, Glob Chang Biol, 24, 2884–2897, https://doi.org/10.1111/gcb.14045, 2018.*
*Berg, A., Sheffield, J., and Milly, P. C. D.: Divergent surface and total soil moisture projections under global warming, Geophys. Res. Lett., 44, 236–244, https://doi.org/10.1002/2016GL071921, 2017.*

*Chadburn, S. E., Burke, E. J., Gallego-Sala, A. V., Smith, N. D., Bret-Harte, M. S., Charman, D. J., Drewer, J., Edgar, C. W., Euskirchen, E. S., Fortuniak, K., Gao, Y., Nakhavali, M., Pawlak, W., Schuur, E. A. G., and Westermann, S.: A new approach to simulate peat accumulation, degradation*

*and stability in a global land surface scheme (JULES vn5.8_accumulate_soil) for northern and temperate peatlands, Geosci. Model Dev., 15, 1633–1657, https://doi.org/10.5194/gmd-15-1633-2022, 2022.*

*Crowther, T. W., van den Hoogen, J., Wan, J., Mayes, M. A., Keiser, A. D., Mo, L., Averill, C., and Maynard, D. S.: The global soil community and its influence on biogeochemistry, Science, 365, eaav0550, https://doi.org/10.1126/science.aav0550, 2019.*

*Davidson, E. A.: Carbon dioxide loss from tropical soils increases on warming, Nature, 584, 198–199, https://doi.org/10.1038/d41586-020-02266-9, 2020.*

*Davidson, E. A. and Janssens, I. A.: Temperature sensitivity of soil carbon decomposition and feedbacks to climate change, Nature, 440, 165, https://doi.org/10.1038/nature04514, 2006.*

*Fan, K., Slater, L., Zhang, Q., Sheffield, J., Gentine, P., Sun, S., and Wu, W.: Climate warming accelerates surface soil moisture drying in the Yellow River Basin, China, J. Hydrol., 615, 128735, https://doi.org/10.1016/j.jhydrol.2022.128735, 2022.*

*Kirschbaum, M. U. F.: The temperature dependence of organic-matter decomposition—still a topic of debate, Soil Biol. Biochem., 38, 2510–2518, https://doi.org/10.1016/j.soilbio.2006.01.030, 2006.*

*Moyano, F. E., Manzoni, S., and Chenu, C.: Responses of soil heterotrophic respiration to moisture availability: An exploration of processes and models, Soil Biol. Biochem., 59, 72–85, https://doi.org/10.1016/j.soilbio.2013.01.002, 2013.*

*Tang, J. and Riley, W. J.: Competitor and substrate sizes and diffusion together define enzymatic depolymerization and microbial substrate uptake rates, Soil Biol. Biochem., 139, 107624, https://doi.org/10.1016/j.soilbio.2019.107624, 2019.*

*Yan, Z., Bond-Lamberty, B., Todd-Brown, K. E., Bailey, V. L., Li, S., Liu, C., and Liu, C.: A moisture function of soil heterotrophic respiration that incorporates microscale processes, Nat. Commun., 9, 2562, https://doi.org/10.1038/s41467-018-04971-6, 2018.*

*Yu, L., Ahrens, B., Wutzler, T., Schrumpf, M., and Zaehle, S.: Jena Soil Model (JSM v1.0; revision 1934): a microbial soil organic carbon model integrated with nitrogen and phosphorus processes, Geosci. Model Dev., 13, 783–803, https://doi.org/10.5194/gmd-13-783-2020, 2020.*

*Yu, L., Caldararu, S., Ahrens, B., Wutzler, T., Schrumpf, M., Helfenstein, J., Pistocchi, C., and Zaehle, S.: Improved representation of phosphorus exchange on soil mineral surfaces reduces estimates of phosphorus limitation in temperate forest ecosystems, Biogeosciences, 20, 57–73, https://doi.org/10.5194/bg-20-57-2023, 2023.*

---

## Author Comment (AC2)

**Response to Anonymous Referee #2**

This paper conducts a series of model experiments using the JSM soil model to explore the response of modeled SOC to warming and drought, with specific focus to depolymerization terms and their temperature sensitivity.

To accomplish this, the authors impose several general warming and drought experiments. I think it would be useful to develop the model experiments following from a specific objective. For example, if the authors want to know how soils are likely to change with climate change in a given location, they may apply one or two warming scenarios using forcings derived from the Shared Socioeconomic Pathways (running the temperature and/or moisture changes over time rather than stepping up by a global average) at the site in Germany. While it is much more work to extract the appropriate forcing information, it gives a much more specific sense of what the model expects, and the potential to evaluate predictions.

There may be other objectives, such as to conduct a sensitivity analysis of JSM model parameters. With this objective, the authors could do a literature survey of possible parameter ranges and explore the SOC response space given systematic combinations of parameter values. Some attention needs to be given to the parameters chosen for analysis. Why were they chosen and not others? What physical or chemical significance do they have?

*We thank reviewer #2 for their comments on our manuscript and helpful suggestions. We apologise for the unclear formulation of the scope and objectives for our study, and aim to improve their descriptions, especially in the introduction. Specifically, our study focuses on modelling the sensitivities of depolymerisation rates to changes in temperature and soil moisture changes. These are processes that have partially been overlooked in dynamic modelling studies before. In this work, we aim to study the effect of these temperature and moisture sensitivities parameterized based on data from a lab study by Allison et al. (2018).*

*For our model study, we do make use of scenario based estimates for climate change (4.5 degree warming by the year 2100 from RCP8.5). While this is different from the by the reviewer suggested warming scenario with gradual temperature increases over a 100 year period, our study does provide insight into the possible direction and magnitudes of change in SOC stocks following a temperature increase. Future soil moisture projections are highly variable in time, space, and very climate model-dependent (Berg et al., 2017), so that selecting one specific forcing dataset for our example site would not provide much new insights beyond what the existing forcing data and drought scenarios provide.*

*The choice of model parameters has been based on field and lab-based literature, and we will describe this more clearly in the methodology section of the paper. Several of these parameters have been tried and tested in the previous model studies by Yu et al. (2020, 2023) and Ahrens et al., (2015, 2020) and the ones that are newly introduced for this study (based on lab results from Allison et al. (2018)) received their own paragraph (2.3) in the methods section. The methodology will be revised to better convey this to the readers.*

To understand how well JSM models the SOC response to temperature or soil moisture, authors could compare model output to data.

*Given the long timeline of the simulations (100 years) and the novel focus of our study on the sensitivities of microbial processes to temperature and drought at different soil depths, we on the one hand see that such data is not readily available and therefore a comparison of the model outputs to observational data falls outside the scope of the study. In line with our reply to reviewer#1, we argue that on the other hand, available respiration data, for example, would only match the first few years of the simulations for the experiments and allow us to verify the bulk flux from the complete soil profile under 'ambient' climatic conditions (and not match the various the warming/drought experiments), and SOC measurements from warming/drought field experiments would be highly impacted by changes in plant productivity (litter inputs). A discussion of this study limitation is already part of the original manuscript. Please note that while JSM is a relatively new model, the processes and parameters for JSM and its predecessor COMISSION have been successfully tested against observations for various applications by Yu et al. (2020, 2023), Ahrens et al. (2015, 2020), and Fleischer et al. (in prep, but see* [https://meetingorganizer.copernicus.org/EGU22/EGU22-11276.html](https://meetingorganizer.copernicus.org/EGU22/EGU22-11276.html)*).*

There are many options and I think this study could be quite interesting if expanded to fit into one of these frameworks, or a different one, as long as there is some clearer justification for the chosen experiments.

*The kindly suggested options by the reviewer, as well as the comments by reviewer #1, indicate we were not successfully relaying the intended scope and objectives for our study in the original manuscript. We apologise for the unclear language and revise to make them more clear, alongside a better description of why the parameter values for the temperature sensitivities of the depolymerisation rates for litter and microbial residues were chosen from the lab-based study by Allison et al. (2018).*

L46: I think there needs to be more description about what Km means in physical terms, and why it is important. As written, it seems a bit arbitrary to explore the temperature sensitive of a fairly abstract parameter in the kinetics equation and not the other sources of temperature sensitivity in the model.

*The half-saturation constant, Km, is the concentration (in this case of microbial biomass, Cb) where the depolymerization rate is 0.5\*Vmax (Eq. 1). Km is not an abstract parameter at all, but an important determinant in the reaction rates of enzyme kinetics: A low half-saturation constant value would mean that the reaction rate is only limited at very low microbial biomass concentrations (e.g. in the subsoil). A high value means that the reaction rate will only be unlimited at very high microbial biomass concentrations (e.g. in the topsoil). The Km parameter for depolymerization can be viewed as the affinity of an enzyme to bind to different polymeric substrates for depolymerization, i.e. plant litter or microbial residues (Tang and Riley, 2019).*

*The value of the half-saturation constant itself is sensitive to temperature and soil moisture and thereby has the potential of further accelerating or counteracting SOC decomposition rates in a warming climate (e.g. see Allison et al., 2018; Davidson and Janssens, 2006; Davidson et al., 2012). To our knowledge, this has not been explored in a dynamic modelling setting before, which makes this study very novel. Other temperature sensitivities, through the maximum reaction rates (Vmax) of the microbial depolymerisation and uptake rates, as well as the sorption and desorption rates, are also active in our study - these maximum*

*reaction rates are also affected by 4.5 degree warming through their respective Q10 values (listed in Table 1). These temperature sensitivities, however, are more well-established in the microbial-mineral SOC model literature (Wang et al., 2012, 2013) than the temperature sensitivities of the Km which were, with the interplay with soil moisture sensitivity, the focus of this study. These warming effects on SOC decomposition are presented and discussed at length in sections 3.1 and 4.1 of the manuscript.*

L193: I found the choices of Q10 parameter experiment a little confusing. Q10=1 and Q10=literature values makes sense, but I didn't quite understand the hypothesis underlying the choice to set both parameters to either litter or residue.

*We apologise for this confusion and will further expand the motivation for this choice in sections 2.3 and 2.4 of the methodology, where we refer to the lab-based measurements of Allison et al. (2018). Setting these values to either all litter or all microbial residues was done to showcase the effects of an overall Q10,KM of 1.3 and a Q10,Km of 0.7. Setting both parameters to either litter or residue was hence not based on a mechanistic hypothesis but rather as an exploration of the edge cases of all Q10,KM below 1 and all Q10,Km above 1. We will make clearer that this was intended as a sensitivity study and model experiment rather than testing a specific hypothesis.*

L223: Rather than visual inspection to determine steady state, you could set some quantitative measure such as <1% change in stock (or a moving average) over the last 100 years.

*We used linear regression to determine the steady-state assumptions and will mention this in the methods.*

L230: Sulman et al. 2018 (https://link.springer.com/article/10.1007/s10533-018-0509-z) demonstrated that different soil C models had widely different assumed temperature sensitivities of mineral associated carbon. Can you make a compelling case that MAOM is less temperature sensitive than microbial processes?

*Yes, in a review article on uncertainty in soil C feedbacks, Bradford et al. (2016) recognize the important role that microbes play in the stabilisation and formation of stabilised SOC, which is less sensitive to warming (Fig. 3 in Bradford et al. (2016), and see Tang and Riley (2015)). We will include these references in the manuscript.*

*The five models analysed in Sulman et al. (2018) contain four models that include microbially mediated decomposition rates (CORPSE, MIMICS, RESOM and MEND), of which only one (the MEND model) has non-linear representations of mineral SOC protection, comparable to JSM, which can be decomposed by microbes (not possible in RESOM). Therefore, the widely-ranged values reported for the other four models reflect the differences in model structure, as each value must be somehow calibrated to fit inside its specific model framework, rather than reflect values which are process-based as is the case in the MEND model and JSM.*

*In our study, we make use of the values reported by Wang et al. (2013) for the temperature sensitivity of sorption and desorption parameters $Q_{10,adsorption}$ and $Q_{10,desorption}$ (Table 1). In this study (for the MEND model), Wang et al. (2013) developed and tested parameter values for*

*explicitly modelling the desorption and adsorption of SOC to mineral surfaces based on literature (reported in Table 3 of Wang et al., 2013). An application for JSM (with its predecessor model, COMISSION) has been successfully reported by Ahrens et al. (2015, 2020).*

L267: Define here which pools you consider to be in POC vs MAOC. I think this may be the first occurrence of the abbreviation so you should define the terms as well.

*We apologise for the late definition in the original manuscript (L.357-358). We will change this and add it to section 2.5.*

Figures 5 seems very similar to Figure 4, and not additive to the effect of temperature shown in Figure 3 – yellow line. Why is that? Does this imply that SOC in JSM is more sensitive to soil moisture than temperature?

*The figures indeed look similar, because the effect of drought and warming on SOC stocks through the half-saturation constant, Km, as shown in Fig. 5 is of a much smaller magnitude than the effect of drought + warming (Fig. 4, without temperature sensitivity of Km). Drought rapidly increases the amount of POC in the topsoil as litter inputs accumulate over the simulation period (Fig. A2, 0–6 cm orange and pink lines), which makes the temperature effect on Km very small. The temperature effect can be better observed in the deep soil layer, as Km becomes more important at lower microbial biomass concentrations (Fig. 1, Eq. 3).*

*We would like to point out that the results shown in Figure 4 also include the effects of soil warming by 4.5 degrees, as is the case in ALL model experiments (but not in the ambient model run, dark blue line Fig. 3). We colour coded the lines in Figs 3 – 5 for easy visual comparison between figures: In Figure 3 and 4, the purple lines are identical, and in Fig. 3 and 5, the yellow lines are identical. We apologise for not clarifying this better in the text and figure captions, and will improve this in the revised version of the manuscript. We could also, for example, revise Figure 5 to only show the differences between these model runs and the ones shown in Fig. 4, to highlight that the effects of Q10,Km are stronger in the subsoil than the topsoil – i.e. visualising the % SOC changes listed in Table 2 and reporting in section 3.3 (L. 290 – 316).*

L334: The temperature sensitivity of adsorption and desorption seems like potentially important parameters.

*Yes, they play a role in explaining the overall lower SOC losses from the subsoil in response to soil warming, as the amount of adsorbed carbon is larger in the subsoil (Fig A1). The MAOC pool is not directly affected by microbial depolymerisation (Fig. 2), but temperature does affect the desorption rates of MAOC into the DOC and microbial residues pool. As a result, the lower temperature sensitivities of adsorption and desorption contribute to the overall lower apparent temperature sensitivity of the total SOC pools when the ratio of MAOC:POC is high. We discuss this in lines 341 – 358, but will mention this important distinction earlier on in the manuscript.*

**References:**

Ahrens, B., Braakhekke, M. C., Guggenberger, G., Schrumpf, M., and Reichstein, M.: Contribution of sorption, DOC transport and microbial interactions to the 14C age of a soil organic carbon profile: Insights from a calibrated process model, Soil Biol. Biochem., 88, 390–402, https://doi.org/10.1016/j.soilbio.2015.06.008, 2015.

Ahrens, B., Guggenberger, G., Rethemeyer, J., John, S., Marschner, B., Heinze, S., Angst, G., Mueller, C. W., Kögel-Knabner, I., Leuschner, C., Hertel, D., Bachmann, J., Reichstein, M., and Schrumpf, M.: Combination of energy limitation and sorption capacity explains 14C depth gradients, Soil Biol. Biochem., 148, 107912, https://doi.org/10.1016/j.soilbio.2020.107912, 2020.

Allison, S. D., Romero-Olivares, A. L., Lu, L., Taylor, J. W., and Treseder, K. K.: Temperature acclimation and adaptation of enzyme physiology in Neurospora discreta, Fungal Ecol., 35, 78–86, https://doi.org/10.1016/j.funeco.2018.07.005, 2018.

Berg, A., Sheffield, J., and Milly, P. C. D.: Divergent surface and total soil moisture projections under global warming, Geophys. Res. Lett., 44, 236–244, https://doi.org/10.1002/2016GL071921, 2017.

Bradford, M. A., Wieder, W. R., Bonan, G. B., Fierer, N., Raymond, P. A., and Crowther, T. W.: Managing uncertainty in soil carbon feedbacks to climate change, Nat. Clim. Change, 6, 751–758, https://doi.org/10.1038/nclimate3071, 2016.

Davidson, E. A. and Janssens, I. A.: Temperature sensitivity of soil carbon decomposition and feedbacks to climate change, Nature, 440, 165, https://doi.org/10.1038/nature04514, 2006.

Davidson, E. A., Sudeep, S., Samantha, S. C., and Savage, K.: The Dual Arrhenius and Michaelis–Menten kinetics model for decomposition of soil organic matter at hourly to seasonal time scales, Glob. Change Biol., 18, 371–384, https://doi.org/doi:10.1111/j.1365-2486.2011.02546.x, 2012.

Sulman, B. N., Moore, J. A. M., Abramoff, R., Averill, C., Kivlin, S., Georgiou, K., Sridhar, B., Hartman, M. D., Wang, G., Wieder, W. R., Bradford, M. A., Luo, Y., Mayes, M. A., Morrison, E., Riley, W. J., Salazar, A., Schimel, J. P., Tang, J., and Classen, A. T.: Multiple models and experiments underscore large uncertainty in soil carbon dynamics, Biogeochemistry, 141, 109–123, https://doi.org/10.1007/s10533-018-0509-z, 2018.

Tang, J. and Riley, W. J.: Weaker soil carbon–climate feedbacks resulting from microbial and abiotic interactions, Nat. Clim. Change, 5, 56–60, https://doi.org/10.1038/nclimate2438, 2015.

Wang, G., Post, W. M., Mayes, M. A., Frerichs, J. T., and Sindhu, J.: Parameter estimation for models of ligninolytic and cellulolytic enzyme kinetics, Soil Biol. Biochem., 48, 28–38, https://doi.org/10.1016/j.soilbio.2012.01.011, 2012.

Wang, G., Post, W. M., and Mayes, M. A.: Development of microbial-enzyme-mediated decomposition model parameters through steady-state and dynamic analyses, Ecol. Appl., 23, 255–272, https://doi.org/10.1890/12-0681.1, 2013.

Yu, L., Ahrens, B., Wutzler, T., Schrumpf, M., and Zaehle, S.: Jena Soil Model (JSM v1.0; revision 1934): a microbial soil organic carbon model integrated with nitrogen and phosphorus processes, Geosci. Model Dev., 13, 783–803, https://doi.org/10.5194/gmd-13-783-2020, 2020.

Yu, L., Caldararu, S., Ahrens, B., Wutzler, T., Schrumpf, M., Helfenstein, J., Pistocchi, C., and Zaehle, S.: Improved representation of phosphorus exchange on soil mineral surfaces reduces estimates of phosphorus limitation in temperate forest ecosystems, Biogeosciences, 20, 57–73, https://doi.org/10.5194/bg-20-57-2023, 2023.

---

## Author Response (AR1)

**Authors' response to anonymous referees #1 and #2**

*We thank the two anonymous reviewers for taking the time to read and comment on our manuscript. Below we provide detailed comments to the reviewers' questions and suggestions, and describe the improvements we made for the revised manuscript.*

**Response to Anonymous Referee #1**

The study by Pallant et al. aims to investigate the interactions between microbial depolymerization and climate change, specifically the influence of temperature and moisture on SOC decomposition and the differential response of soil layers to warming and drought, using the JSM model. Despite the study's promising title and significant potential contributions to future SOC research, it falls short in several key areas, preventing it from being a comprehensive study. In its present state, it seems incomplete, suitable at best for a sensitivity analysis within a broader, more thorough investigation. I recommend withholding publication until substantial improvements are made.

*We apologise for the apparent mismatch between the title and the content, and adjusted the manuscript's title. Additionally, the updated manuscript includes clearer descriptions of the study's objectives, novelty and scope, in line with our response to the comments and suggestions raised by anonymous reviewer #2. We realise that many of the issues raised by the reviewer below were caused by an incomplete description of some of the model processes, where we assumed that in-text references to previously published manuscripts would suffice. We apologise for this confusion and made substantial changes to the text: The methodology section was revised, in particular with regard to our choice of model parameter values as well as the vertical process representation in JSM and the SOC decomposition processes influenced by soil moisture and temperature along a vertical gradient. We also rewrote the introduction to better outline the study's scope and objectives. Overall, we made these changes to increase the comprehensiveness of the study, and improve the manuscript's readability and understandability to the journal's readers.*

**General comments:**

The title's implication of a drought-focused study misaligns with the content, which predominantly consists of sensitivity tests devoid of any empirical assessment. The absence of observational data reduces the study to hypothetical model outputs, with an overemphasis on parameter sensitivity to temperature and moisture.

*We regret the apparent mismatch between the title and the content, and adjusted the title to "Modelling the impact of climate-substrate interactions on soil organic matter decomposition with the Jena Soil Model" so that it better reflects the paper's scope.*

*This is indeed a modelling study with a strong focus on the sensitivities of depolymerisation rates to changes in temperature and soil moisture changes. These are processes that have partially been overlooked in dynamic modelling studies before, in particular the soil moisture impact. In this work, we investigate the effect of these temperature and moisture sensitivities parameterised based on data from a lab study by Allison et al. (2018). While JSM is a relatively new model, the processes and parameters for JSM and its predecessor*

*COMISSION have been successfully tested against observations for various applications by Yu et al. (2020, 2023), Ahrens et al. (2015, 2020), and Fleischer et al. (in prep, but see https://meetingorganizer.copernicus.org/EGU22/EGU22-11276.html). Making frequent reference to these existing studies, we did not conduct additional comparisons between model outputs and observational data in this study. Our main reason to do so is a lack of suitable data: e.g. available respiration data would only cover the first few years of the simulations for the experiments, and SOC measurements from warming/drought experiments would be highly impacted by changes in plant productivity (litter inputs). A discussion of this study limitation was already part of the original manuscript at the end of section 4.4, which was further extended by mentioning the competition between plants and microbes for resources, through e.g. changes over time in plant litter inputs and microbial CUE.*

This study also overlooks other crucial dynamics in SOC, such as diffusion and advection, which could significantly influence SOC distribution within soil columns and thus its response to soil temperature and moisture changes. The omission of factors like soil texture and vegetation dynamics further limits the study's scope.

*We disagree on that point: JSM as it was applied in this study already includes the above-mentioned processes, which are described in Ahrens et al. (2015, 2020) and Yu et al. (2020). Diffusion, advection, and soil texture are all considered by the model. Soil texture influences soil moisture and water fluxes but also the sorption of DOM and microbial residues via the maximum sorption capacity, Qmax. For the sake of brevity, these are not described in detail in this manuscript, but we mention them in the methodology section of the updated manuscript with the appropriate references to their process description with equations. The aim of our study is to investigate in detail the effects of interacting temperature and moisture sensitivities in a dynamic Michaelis-Menten term. To our knowledge, this has not been done before in a comprehensive manner and this warrants an in-depth modelling study. This novel aspect of our study is better highlighted in the thoroughly revised introduction of the updated manuscript.*

*To fit the objectives for this study, we made the conscious choice to not have an active response of the vegetation to moisture and temperature changes and use constant litter forcing over time, to be able to pinpoint the changes in modelled SOC dynamics in response to the climatic drivers in isolation from those driven by changes in litter inputs.*

The introduction should be revised to avoid methodological details and instead provide a comprehensive overview of the factors controlling SOC dynamics. Additionally, the methodology does not adequately discuss SOC discretization or the interplay between the topsoil and subsoil layers.

*We feel that a comprehensive overview of the factors controlling SOC dynamics falls beyond the scope of the introduction, since we want to focus on the interplay of soil temperature and soil moisture sensitivities in a microbial-mineral model at different soil depths. We discuss the various processes that are regulated by temperature and moisture, with a strong emphasis on microbial depolymerisation rates and reverse Michaelis-Menten kinetics. The introduction now specifically covers the different temperature and moisture sensitivities of microbial decomposition processes, the role of organo-mineral interactions, and*

*depth-specific processes. We refer to previously published work with JSM for other controlling factors of SOC decomposition that are beyond the temperature and moisture sensitivity scope of this manuscript.*

*We very much agree with the reviewer that the methodology section should have included a more thorough description of the interplay between top and subsoil layers. We added this information and renamed Section 2.2 to "Vertical process representation in the Jena Soil Model". Additionally, the descriptions of SOC advection and diffusion mentioned in the previous comment were included in the methodology.*

I have confined my comments to the methodology section because the paper's foundation needs strengthening before the results and discussion can be meaningful.

*We thoroughly revised the introduction and methodology sections to incorporate the above comments to improve the paper's foundation, especially with regard to the scope of this study and the JSM model process descriptions.*

**Specific comments:**

L20: Clarify the term "long-term" to provide proper context.

*We changed to "We find that soil warming leads to SOC losses at a timescale of a century"*

L20: Define "SOC-specific Q10" to elucidate its relevance to the study.

*We agree, and the updated abstract no longer contains this text. Instead, we use a more general description, referring to specific $Q10_{Km}$ values associated with different enzymes for the breakdown of litter or microbial residues. These two Q10 values (below 1 (0.7), and above 1 (1.3)) have opposing effects on decomposition and interact with the amount of microbial biomass via the Michaelis-Menten term. This introduced a different sensitivity between topsoil and subsoil as microbial biomass decreases with the amount of root litter inputs and advective and diffusive transport inputs which decrease with depth.*

L21: Distinguish between "reduce" and "accelerated" processes to prevent ambiguity.

*The revised abstract no longer contains this sentence. Instead, we write "While absolute SOC losses were highest in the topsoil, we found that the temperature and moisture sensitivities of Km were important drivers for SOC losses in the subsoil – where microbial biomass is low and mineral-associated OC is high."*

L23: Replace "SOC gain" with "SOC accumulation" to accurately reflect changes in SOC cycling during different environmental conditions.

*We agree and rephrased to "SOC accumulation" in the whole manuscript.*

L24: When using the term "decomposition rate," additional details should be provided for clarity. It might be preferable to use "SOC loss" for simplicity and better understanding.

*We rephrased to SOC balance instead of SOC loss, since it can be both a SOC gain or loss.*

L25-26: You need to be clearer. Maintain consistency in the comparison of temperature and moisture effects on both topsoil and subsoil.

*We rephrased to "In this study, we show that while absolute SOC changes driven by soil warming and drought are highest in the topsoil, SOC in the subsoil is more sensitive to warming and drought due to the intricate interplay between K_m, temperature, soil moisture, and mineral-associated SOC"*

L28-29: These assertions require supporting references for validation

*We added the missing references and included Fan et al. (2020), Crowther et al. (2019) in the first part, as well as Davidson (2020) and Kirschbaum (2006) in the second part.*

L34-35: The phrase "most important" is subjective and should be reworded to reflect that soil temperature and moisture are significant controlling factors. This section would benefit from an expanded discussion on the key factors that regulate soil dynamics.

*The wording might be perceived as subjective and we rephrased to "Soil temperature and soil moisture are two primary controlling factors of microbial decomposition rates, and thereby the carbon turnover rate of soils (Davidson and Janssens, 2006; Moyano et al., 2013; Yan et al., 2018)."*

*Additionally, we put higher emphasis on the importance of SOC stabilisation on mineral surfaces as another important factor determining the fate of SOC stocks (Ahrens et al., 2020; Dwivedi et al., 2017; Sokol et al., 2022), and discuss the differences between top and subsoil layers. Overall, the revised introduction discusses the most important factors that fall within the scope of this study.*

L35: Clarify the term "de(stabilisation)" for better understanding.

*This sentence is no longer part of the revised introduction.*

L37-39: Reformulate these lines for enhanced clarity and readability.

*We apologise for unclear wording and rephrased to "Given the importance of SOC stocks and their sensitivities to climate change, a better understanding and representation of these complex interactions in coupled C cycle-climate models is extremely important for a better understanding of the carbon-climate feedback (Todd-Brown et al., 2014)."*

L41: Specify the models referred to in this line.

*This sentence is no longer part of the revised introduction.*

L43-80: Shift the detailed methodological exposition to the methods section, using the introduction to provide a broader overview.

*We thoroughly revised this part of the introduction and eliminated the equations from the text to make it more understandable and less technical. To our knowledge, previous works have primarily focussed on the temperature sensitivities of Vmax (or the apparent sensitivities of (heterotrophic) respiration rates). The novelty of our study is the focus on the temperature*

*and moisture sensitivities on depolymerisation rates through the half-saturation constant (Km), which to our knowledge has not been shown in a modelling study before. Reverse Michaelis-Menten kinetics (compared to forward kinetics) is less commonly used in models, and may be less known to the readers of Biogeosciences. Therefore, we think the conceptual figure and text help to clarify why microbial depolymerisation rates can both increase or decrease in response to soil warming and drought/soil wetting through Km, especially at low microbial biomass concentrations.*

L83: A reference is missing and should be included.

*This sentence is no longer part of the revised introduction.*

L88: What about peatlands? Inclusion or exclusion of them would significantly change your findings. It is important to note that the moisture function, decomposition term, and vertical dynamics of SOC distinctly vary in wetland environments.

*This sentence is no longer part of the revised introduction. But we agree with the reviewer that this is definitely true for peatlands, and the traditional empirical model approaches we refer to in the introduction also don't describe SOC-moisture-temperature dynamics in these ecosystems very well (or even do consider them at all, see e.g. Chadburn et al. (2022)). Our results are for a mineral soil profile, so we feel that explicitly discussing peatlands in the introduction falls outside of the scope. Peatlands and their different moisture dynamics, however, are briefly discussed at the end of Section 4.4.*

L95-99: Avoid repeating information from the methods section to streamline the text.

*We removed the redundant information on JSM.*

L113: Incorporate the previously mentioned introductory information here for coherence.

*As mentioned in the earlier comments above, we included additional information for JSM on the processes of diffusion, advection, and soil texture. We made textual edits to the introduction and methodology sections, and updated Fig. 2 and its caption (see comment below on L116).*

L115: Briefly explain each parameter, including those in Table 1, and cite the source of the formulations. Additionally, explain the rational for such formulation for DOC?

*The parameters and constants from Table 1 are all explained in the text, with references in the text and Table 1.*

*The formulation of Eq. 1 describes the depolymerisation of litter and microbes to DOC. Hence, it is not a formulation for DOC. JSM is based on the COMISSION model, so we included the references to this model by adding Ahrens et al. (2015, 2020). The reverse Michaelis-Menten kinetics formulation for depolymerisation is based on the derivations by Tang and Riley (2019) that we mentioned in the previous sentence. We apologise for the unclear language and rephrased to "The depolymerisation rate ($mmol\ C\ m^{-3}\ hour^{-1}$) of litter or microbial residues into the dissolved organic carbon (DOC) pool (following Ahrens et al., 2015, 2020) is described as:"*

Eq1: For this and all subsequent equations, it is necessary to specify the units of the calculated flux. Additionally, clarify if any conversions were performed to derive these units from the parameters listed in Table 1.

*We added the units of all fluxes (Eqs. 1 and 5) and apologise for some small typesetting mistakes we found and corrected in the superscript of the units in Table 1. The temperature conversions of $Q_{10,Km,P}$ and $Q_{10,Km,R}$ (Allison et al., 2018) from the lab study's reference temperature of 16 °C to JSM's reference temperature of 20 °C is described in the paper using Eq. 3, and separately marked in Table 1 with an asterisk. No other conversions were made. The model itself runs on a half-hourly timestep and all units in the model code are defined uniformly using SI units (mol, K, seconds). For the sake of clarity (as these e.g. maximum depolymerisation rates or microbial growth rates are commonly measured in other time units (e.g. days, hours) or concentrations (mol, or mmol per soil volume)), and easy cross-referencing, however, we chose to report the values in Table 1 as they were presented in the original model description papers by Ahrens et al., 2020 and Yu et al., 2020. We added this information to the caption below Table 1.*

L116: Although the layered structure of the soil has been mentioned and is depicted in Figure 2, there is a noticeable absence of any depth indicators in the equations or in the estimations provided.

*We added this clarification in the methods section in section 2.2 "Vertical process representation in JSM", to explain that all C pools and forcings are functions of time and depth. In addition, the caption of Fig. 2 was changed so that it clearly mentions that all soil layers (the grey cylinders) contain all C pools (black boxes) and C fluxes (black arrows).*

Figure 2: The figure needs to be revised for clarity, including an explanation of the brown arrows and a clear differentiation between soil pools and depth indicators.

*The brown arrows represent the above and belowground inputs. We revised Fig. 2 and separated these labels into 'aboveground litter' and 'root litter', and moved the label for the belowground arrows (root litter inputs) to the left. Every model pool is present in each soil depth. We clarify this in the updated caption of Fig. 2 (see previous comment).*

Table 1: Complete the table by labelling parameters that lack units as "unitless." Additionally, is there a reference for the parameter R that can be provided?

*R, the universal gas constant, is not a parameter but a well-known natural constant, as mentioned under Eq. 3, L.173. Since the 2019 redefinition of SI base units, it has an exact numerical value with zero uncertainty, and we report the universal gas constant in $J\ K^{-1}\ mol^{-1}$, adhering to the journal's use of standard SI units. A reference would be [https://en.wikipedia.org/wiki/Gas_constant](https://en.wikipedia.org/wiki/Gas_constant). Because R is such a well-known constant, we removed it from Table 1 to avoid further confusion. We added '–' to Table 1 for unitless variables where applicable, and also updated the Table heading from "Parameter" to "Parameter/constant" for clarity.*

L145: The nutrient dynamics have a significant influence on uptake and other soil processes. While you state that the model accounts for nutrients, there's a missed opportunity to thoroughly explore and discuss this aspect within the current study.

*We focus on one novel aspect in this study: the interaction of soil moisture and soil temperature sensitivity of a dynamic Michaelis-Menten term. Our focus on the C cycle with the temperature sensitivities of different litter types, as well as the various warming and drought experiments already reveals a large range of complex interactions between microbes, organo-mineral interactions, and climatic variables within this dynamic modelling system. Extending this work by including nutrient dynamics and its temperature and moisture sensitivity leads to a path far outside the domain covered by our study's model approach. At the end of the discussion of the updated version of this manuscript we now mention the inclusion of nutrient dynamics among the future research directions with novel modelling frameworks such as JSM.*

L147: Consideration of root distribution across various soil depths is missing. An analysis of how roots are spread throughout the soil profile could offer valuable insights into soil dynamics.

*We apologise for not clearly communicating that we consider root distribution across various root depths. Figure 2 and sections 2.1 and 2.2 of the manuscript were revised accordingly. The root distribution is the main factor for different microbial biomass levels in different depths (Ahrens et al., 2020). We included this important information and its reference in section 2.2 of the revised manuscript.*

L153: Include the sorption/desorption equation to complete the methodological description.

*Sorption and desorption are clearly described in Ahrens et al. (2015, 2020), which we refer to in L.187-191 and L.207 - 212 of the updated manuscript. While checking this sentence we found and corrected two additional small mistakes:*
*1) In the description of the activation energies, all Q10 values used in this study are based on literature, and not calibrated.*
*2) Lines 155 – 158 of the original manuscript are the Caption of Table 1. We corrected this formatting error and moved this text back below Table 1.*

L184: What do you mean by first model run? Explain the rationale behind the first model run and the selection of a 100-year period.

*We rephrased this section by describing this model run after the spin-up as the ambient model run, to be consistent with the rest of the manuscript. The selection of a 100-year simulation period was done to allow the different SOC pools to return to steady state after the respective disturbances (warming, drought) so that the changes in SOC stocks between the start and the end of the experiments could be calculated (Table 2).*

L185: The decision to focus solely on the 0-50 cm soil layer raises questions. Should processes like advection or diffusion be considered, the entire soil column to its maximum depth would likely experience changes during the spin-up period. Why then, is the analysis limited to this specific depth range?

*The processes of advection of DOC and diffusion to represent bioturbation are considered in JSM. We added this information to section 2.1 (L.135 - 136). In JSM, there are 15 soil layers, of which 6 layers between 0 and 50 cm depth, and we made this more clear in the methodology section. These 6 layers experience the strongest influence of transport. Soil*

*layer thickness increases with increasing soil depth, at the following depth intervals: 0 – 6, 6 – 13, 13 – 20, 20 – 26, 26 – 36, and 36 – 50 cm. So we did not focus on only one layer between 0 – 50 cm, but compared the processes in the vertical profile setting with multiple soil layers. By comparing a shallow and a deeper layer, we were able to look at the differences along a gradient of microbial activity – the majority of microbial activity and the resulting heterotrophic respiration flux takes place in the upper soil layers. Additionally, the difference between the shallow and deeper layer illustrate the effect of decreasing root litter inputs with depth.*

*JSM also provides model outputs below 50 cm, which were checked but not shown in the manuscript. These deeper soil layers indeed take many years to reach equilibrium, as the reviewer points out. So another, more practical reason for choosing to focus on the layers until 50 cm depth is that this significantly reduces the necessary model spin-up time and thereby the use of computational resources. Despite these layers not being in steady state after spinup period, we did check the patterns in the layers between 50 and 100 cm depth and found they are similar to what we show in the manuscript. We also found that for the drought and warming experiments, these deeper layers needed more than 100 simulation years to reach a new steady state after disturbance, but they did eventually reach it.*

*The model's vertical process representation and inputs are described more clearly in the methodology section of the revised manuscript (Section 2.2). We also added the description of the model's steady state check after the spinup period using linear regression.*

L186: Detail the approach used to assess steady-state conditions.

*We used a linear regression model to check the steady state assumptions and mention this in the methods of the revised manuscript. It is also visually very clear from the model results, e.g. in Figure 3.*

L189: Clarification is needed on the term "all ambient soil temperature." Does this refer to temperatures at every soil depth? Moreover, to isolate the impact of temperature alone, it would be necessary to perform simulations with varying temperature and moisture, and then compare these to simulations with only temperature variation to accurately determine temperature sensitivity.

*We apologise for unclear wording. Temperatures are depth-specific, which we better clarified in the methods section (2.2). We have improved the description of how the depth-specific litter, temperature and soil moisture data were generated in section 2.2, which now explicitly states: "The soil moisture and soil temperature forcing data from 2000 - 2012 are referred to as ambient soil moisture and soil temperature data from this point."*

*These depth-specific temperatures were increased by 4.5 K for each depth. Similarly, the depth-specific moisture values were decreased by 10, 20 or 30% in the different drought experiments. We have improved the description of the different warming and drought experiments in section 2.4.*

*We did consider the isolated effects of soil warming or drought by including model experiments where either temperature Fig. 3, blue line) or soil moisture (Figs. 3 & 4, purple line; Fig. 5, yellow line) was kept at its original, depth-specific input values.*

L194-196: Justify the chosen experimental values.

*The full justification for these values was provided in the original manuscript in lines 166-183, under a separate section of the manuscript titled "2.3 Choice of Q10,Km values for polymeric litter and microbial residues". In short, the values are based on a lab study on the temperature sensitivities for different enzymes involved in the breakdown of soil carbon substrates (Allison et al., 2018). The Q10,Km value for the depolymerisation of litter was based on measurements for the enzymes β-xylosidase and total oxidase, and the Q10,Km value for the depolymerisation of microbial residues was based on measurements for the enzyme leucine aminopeptidase.*

L201: What was the rational for 10% changes and why only between 0.9-0.7 values?

*We apologise for not clarifying this in the methodology section. Part of the rationale was somewhat hidden in the introduction, and we now clearly state the rationale in Section 2.4 of the revised manuscript. We used the following rationale (L. 258 - 267): "Soil drying is expected for most of the globe (Wang et al., 2022b and references therein), but drought intensity is uncertain and may vary locally (Cook et al., 2020; Hsu and Dirmeyer, 2023). Soil moisture change projections are very uncertain: 60 projected global lateral and vertical distributions of future soil moisture were highly diverse both in their predicted spatial and vertical distributions (Berg et al., 2017). The multi-model mean of this study showed reductions in subsurface and deep surface soil moisture up to 30% by the year 2100. Given the large divergence between these model projections, we chose to simulate drought by reducing the depth-specific soil moisture values from the forcing dataset in steps of 10%, to be able to compare the effects of a relatively mild versus increasingly stronger droughts on SOC stocks."*

L210: Confirm whether equilibrium was reached after 100 spins and discuss mass balance and steady-state conditions for each run.

*We used a linear regression model to check the steady state assumptions and expand the text describing the steady-state check in the methodology (section 2.4). Mass balance checks are performed in the QUINCY-JSM model code, and for every run we have log files showing that the mass balance is closed. Furthermore, the QUINCY-JSM model is part of the EucFACE-MIP data-model comparison project, and a reference to the detailed descriptions of the mass balance checks for C, N and P are recently published in Jiang et al. 2024).*

L216: The mention of R packages is less critical than a thorough analysis of the outputs. Consider focusing on the methods to output evaluation here.

*We expanded this section with a short description of the smoothed curve shown in Figures 3-5 which visually supports the data points plotted for the 100 simulation years. We strongly feel that the mentioning of R packages with citations are important and should remain part of the manuscript, as we are grateful to the many (scientific) open source software developers for their important contributions, and acknowledge this by crediting their work with citations.*

**Response to Anonymous Referee #2**

This paper conducts a series of model experiments using the JSM soil model to explore the response of modeled SOC to warming and drought, with specific focus to depolymerization terms and their temperature sensitivity.

To accomplish this, the authors impose several general warming and drought experiments. I think it would be useful to develop the model experiments following from a specific objective. For example, if the authors want to know how soils are likely to change with climate change in a given location, they may apply one or two warming scenarios using forcings derived from the Shared Socioeconomic Pathways (running the temperature and/or moisture changes over time rather than stepping up by a global average) at the site in Germany. While it is much more work to extract the appropriate forcing information, it gives a much more specific sense of what the model expects, and the potential to evaluate predictions.

There may be other objectives, such as to conduct a sensitivity analysis of JSM model parameters. With this objective, the authors could do a literature survey of possible parameter ranges and explore the SOC response space given systematic combinations of parameter values. Some attention needs to be given to the parameters chosen for analysis. Why were they chosen and not others? What physical or chemical significance do they have?

*We thank reviewer #2 for their comments on our manuscript and helpful suggestions. We apologise for the unclear formulation of the scope and objectives for our study, and aimed to improve their descriptions, especially in the introduction. Specifically, our study focuses on modelling the sensitivities of depolymerisation rates to changes in temperature and soil moisture changes. These are processes that have partially been overlooked in dynamic modelling studies before. In this work, we aim to study the effect of these temperature and moisture sensitivities parameterised based on data from a lab study by Allison et al. (2018).*

*For our model study, we do make use of scenario based estimates for climate change (4.5 degree warming by the year 2100 from RCP8.5, Soong et al. 2020). While this is different from the by the reviewer suggested warming scenario with gradual temperature increases over a 100 year period, our study does provide insight into the possible direction and magnitudes of change in SOC stocks following a temperature increase. Future soil moisture projections are highly variable in time, space, and very climate model-dependent (Berg et al., 2017), so that selecting one specific forcing dataset for our example site would not provide much new insights beyond what the existing forcing data and drought scenarios provide.*

*The choice of model parameters has been based on field and lab-based literature, and we describe this more clearly now in the methodology section of the revised paper. Several of these parameters have been tried and tested in the previous model studies by Yu et al. (2020, 2023) and Ahrens et al., (2015, 2020) and the ones that are newly introduced for this study (based on lab results from Allison et al. (2018)) received their own paragraph (2.3) in the methods section. The methodology was revised to better convey this to the readers.*

To understand how well JSM models the SOC response to temperature or soil moisture, authors could compare model output to data.

*Given the long timeline of the simulations (100 years) and the novel focus of our study on the sensitivities of microbial processes to temperature and drought at different soil depths, we on the one hand see that such data are not readily available and therefore a comparison of the model outputs to observational data falls outside the scope of the study. In line with our reply to reviewer#1, we argue that on the other hand, available respiration data, for example, would only match the first few years of the simulations for the experiments and allow us to verify the bulk flux from the complete soil profile under 'ambient' climatic conditions (and not match the various the warming/drought experiments), and SOC measurements from warming/drought field experiments would be highly impacted by changes in plant productivity (litter inputs). A discussion of this study limitation is already part of the original manuscript. Please note that while JSM is a relatively new model, the processes and parameters for JSM and its predecessor COMISSION have been successfully tested against observations for various applications by Yu et al. (2020, 2023), Ahrens et al. (2015, 2020), and Fleischer et al. (in prep, but see*
*[https://meetingorganizer.copernicus.org/EGU22/EGU22-11276.html](https://meetingorganizer.copernicus.org/EGU22/EGU22-11276.html)).*

There are many options and I think this study could be quite interesting if expanded to fit into one of these frameworks, or a different one, as long as there is some clearer justification for the chosen experiments.

*The kindly suggested options by the reviewer, as well as the comments by reviewer #1, indicate we were not successfully relaying the intended scope and objectives for our study in the original manuscript. We apologise for the unclear language and thoroughly revised the introduction to make them more clear, alongside a better description in the methodology section explaining why these parameter values for the temperature sensitivities of the depolymerisation rates for litter and microbial residues were chosen from the lab-based study by Allison et al. (2018).*

L46: I think there needs to be more description about what Km means in physical terms, and why it is important. As written, it seems a bit arbitrary to explore the temperature sensitive of a fairly abstract parameter in the kinetics equation and not the other sources of temperature sensitivity in the model.

*The half-saturation constant, Km, is not an abstract parameter at all, but describes the substrate binding affinity of enzymes to different substrates for microbial depolymerisation, i.e. plant litter or microbial residues (Tang and Riley, 2019). We agree with the reviewer that this was not explained well in the introduction, and have included a better description of Km in the revised introduction.*

*Km is also the concentration (in this case of microbial biomass, Cb) where the depolymerisation rate is 0.5\*Vmax (Fig. 1). Therefore, it is an important determinant in the reaction rates of enzyme kinetics: A low half-saturation constant value would mean that the reaction rate is only limited at very low microbial biomass concentrations (e.g. in the subsoil). A high value means that the reaction rate will only be unlimited at very high microbial biomass concentrations (e.g. in the topsoil).*

*The value of the half-saturation constant itself is sensitive to temperature and soil moisture and thereby has the potential of further accelerating or counteracting SOC decomposition rates in a warming climate (e.g. see Allison et al., 2018; Davidson and Janssens, 2006;*

*Davidson et al., 2012). To our knowledge, this has not been explored in a dynamic modelling setting before, which makes this study very novel. We have put higher emphasis on this in the completely revised introduction.*

*Other temperature sensitivities, through the maximum reaction rates (Vmax) of the microbial depolymerisation and uptake rates, as well as the sorption and desorption rates, are also active in our study - these maximum reaction rates are also affected by 4.5 degree warming through their respective Q10 values (listed in Table 1). These temperature sensitivities, however, are more well-established in the microbial-mineral SOC model literature (Wang et al., 2012, 2013) than the temperature sensitivities of the Km which were, with the interplay with soil moisture sensitivity, the focus of this study. These warming effects on SOC decomposition are presented and discussed at length in sections 3.1 and 4.1 of the manuscript.*

L193: I found the choices of Q10 parameter experiment a little confusing. Q10=1 and Q10=literature values makes sense, but I didn't quite understand the hypothesis underlying the choice to set both parameters to either litter or residue.

*We apologise for this confusion and expanded the motivation for this choice in sections 2.3 and 2.4 of the methodology, where we refer to the lab-based measurements of Allison et al. (2018). Setting these values to either all litter or all microbial residues was done to showcase the effects of an overall Q10,KM of 1.3 and a Q10,Km of 0.7. Setting both parameters to either litter or residue was hence not based on a mechanistic hypothesis but rather as an exploration of the edge cases of all Q10,KM below 1 and all Q10,Km above 1. The effects of temperature sensitivity of Km have not been investigated in a dynamic modeling study before, and also the use of a negative temperature sensitivity of Km has not been investigated before, which is why we explore these different feedback directions in the paper. In the methods section, we have made clearer that this was intended as a sensitivity study and model experiment rather than testing a specific hypothesis.*

L223: Rather than visual inspection to determine steady state, you could set some quantitative measure such as <1% change in stock (or a moving average) over the last 100 years.

*We used linear regression to determine the steady-state assumptions and now mention this in the methods of the updated manuscript.*

L230: Sulman et al. 2018 (https://link.springer.com/article/10.1007/s10533-018-0509-z) demonstrated that different soil C models had widely different assumed temperature sensitivities of mineral associated carbon. Can you make a compelling case that MAOM is less temperature sensitive than microbial processes?

*Yes, in a review article on uncertainty in soil C feedbacks, Bradford et al. (2016) recognise the important role that microbes play in the stabilisation and formation of stabilised SOC, which is less sensitive to warming (Fig. 3 in Bradford et al. (2016), and see Tang and Riley (2015)). We discuss this in Section 4.1, and included these references in the revised manuscript.*

*The five models analysed in Sulman et al. (2018) contain four models that include microbially mediated decomposition rates (CORPSE, MIMICS, RESOM and MEND), of which only one (the MEND model) has non-linear representations of mineral SOC protection, comparable to JSM, which can be decomposed by microbes (not possible in RESOM). Therefore, the widely-ranged values reported for the other four models reflect the differences in model structure, as each value must be somehow calibrated to fit inside its specific model framework, rather than reflect values which are process-based as is the case in the MEND model and JSM.*

*In our study, we make use of the values reported by Wang et al. (2013) for the temperature sensitivity of sorption and desorption parameters $Q_{10,adsorption}$ and $Q_{10,desorption}$ (Table 1). In this study (for the MEND model), Wang et al. (2013) developed and tested parameter values for explicitly modelling the desorption and adsorption of SOC to mineral surfaces based on literature (reported in Table 3 of Wang et al., 2013). An application for JSM (with its predecessor model, COMISSION) has been successfully reported by Ahrens et al. (2020). We updated the description of sorption and desorption and its temperature sensitivity in the methodology section 2.1, so that it clearly reflects that these parameters are literature-based and have been successfully applied in earlier versions of this model for studies that focused on organo-mineral interactions (Ahrens et al., 2020).*

L267: Define here which pools you consider to be in POC vs MAOC. I think this may be the first occurrence of the abbreviation so you should define the terms as well.

*We agree and apologise for the very late definition in the original manuscript. We improved this by introducing the abbreviations, and which C pools are considered for MAOC and POC in section 2.1 of the methods.*

Figures 5 seems very similar to Figure 4, and not additive to the effect of temperature shown in Figure 3 – yellow line. Why is that? Does this imply that SOC in JSM is more sensitive to soil moisture than temperature?

*The figures indeed look similar, because the effect of drought and warming on SOC stocks through the half-saturation constant, Km, as shown in Fig. 5 is of a much smaller magnitude than the effect of drought + warming (Fig. 4, without temperature sensitivity of Km). Drought rapidly increases the amount of POC in the topsoil as litter inputs accumulate over the simulation period (Fig. A2, 0–6 cm orange and pink lines), which makes the temperature effect on Km very small. The temperature effect through the half-saturation constant can be better observed in the deep soil layer, as Km becomes more important at lower microbial biomass concentrations (Fig. 1, Eq. 1). We added this discussion point to section 4.3 of the revised manuscript (L.490 - 496).*

*Additionally, we would like to point out that the results shown in Figure 4 also include the effects of soil warming by 4.5 degrees, as is the case in ALL model experiments (but not in the ambient model run, dark blue line Fig. 3). We colour coded the lines in Figs 3 – 5 for easy visual comparison between figures: In Figure 3 and 4, the purple lines are identical, and in Fig. 3 and 5, the yellow lines are identical. We apologise for not clarifying this better in the text and figure captions, and included this important information in the captions of Figs. 3 - 5 and the results section of the revised manuscript.*

L334: The temperature sensitivity of adsorption and desorption seems like potentially important parameters.

*Yes, they play a role in explaining the overall lower SOC losses from the subsoil in response to soil warming, as the amount of adsorbed carbon is larger in the subsoil (Fig. A1). The MAOC pool is not directly affected by microbial depolymerisation (Fig. 2), but temperature does affect the desorption and adsorption rates between MAOC and the DOC and microbial residues pool. Compared to the temperature sensitivities of the maximum depolymerisation rates of litter/residues ($Q_{10}$ of 2.16) and microbial uptake ($Q_{10}$ 1.98)), the temperature sensitivities of adsorption ($Q_{10}$ of 1.08) and desorption ($Q_{10}$ of 1.34) are very low. These literature-based $Q_{10}$ values reflect the current scientific understanding that MAOM is less temperature sensitive than POC (Bradford et al. 2016, Tang and Riley, 2015). As a result, the lower temperature sensitivities of adsorption and desorption contribute to the overall lower apparent temperature sensitivity we observed for the total SOC pools when the ratio of MAOC:POC is high (i.e., in the subsoil). We discuss this at length in L.397 – 407 of the revised manuscript and added the references to Bradford et al. 2016; Tang and Riley, 2015). In addition, we mention the literature-based temperature sensitivities of sorption and desorption in the methodology section (L.187 - 188).*

*References:*

[revised manuscript text omitted]

---

## Author Response (AR2)

**Response to Anonymous referee #1**

The authors have incorporated most of the suggestions in this review. However, after reviewing the tracked changes and the responses, several items still need further clarification.

We thank the reviewer for their comments and acknowledging our efforts to improve the manuscript during the previous round of revisions. We have carefully reviewed and responded to the reviewer's additional feedback, summarized item-by-item below. We hope our responses and manuscript improvements will meet the reviewer's expectations.

- The comment on lack of the model comparison against the observational data has not been addressed. Although the authors state that model experiments are based on laboratory studies, they do not include these studies.

We would like to clarify that only the $Q_{10}$ values for the Km of depolymerization are based on the laboratory study by Allison et al. (2018). We dedicated a complete section of the manuscript (Section 2.3) explaining the choice and rationale for these chosen parameter values. The laboratory data from these enzyme assays are published by Allison et al. (2018). Since these results are not generated by us, we feel it is inappropriate to report them in our manuscript beyond what is already written in Section 2.3.

If the reviewer uses the term "model experiments" to refer to the temperature and moisture experiments, we clarify that we did not state anywhere that these are based on laboratory studies, but on findings reported by Soong et al. (2020) and Berg et al. (2017). The different model experiments for our study are summarised in Table 2, and in the manuscript we clearly motivate the choices for the temperature and moisture changes:

- For temperature (L. 255 – 257): "*We chose a 4.5 K step increase for soil warming, because soils, including the deep soil up to 1m, are expected to warm by 4.5 K by the end of the century under representative concentration pathway (RCP) 8.5 (Soong et al., 2020).*"
- For soil moisture (L. 268 – 277): "*Soil drying is expected for most of the globe (Wang et al., 2022b and references therein), but drought intensity is uncertain and may vary locally (Cook et al., 2020; Hsu and Dirmeyer, 2023). Soil moisture change projections are very uncertain: 60 projected global lateral and vertical distributions of future soil moisture were highly diverse in their predicted lateral and vertical distributions (Berg et al., 2017). The multi-model mean of this study showed reductions in subsurface and deep surface soil moisture up to 30% by the year 2100. Given the large divergence between these model projections, we chose to simulate drought by reducing the depth-specific soil moisture values from the forcing dataset in steps of 10%, to be able to compare the effects of a relatively mild versus increasingly stronger droughts on SOC stocks. Specifically, we compared three drought scenarios, where the model's ambient SM inputs are reduced by 10%: Each ambient SM value is multiplied by 0.9, 0.8 or 0.7, respectively (Table 2). As with the warming experiment, the original seasonality in the ambient SM input values is kept intact.*"

Furthermore, independent of prior model evaluations, incorporating such comparisons in the current study would elucidate any improvements or drawbacks of the model relative to

earlier versions and observational data. I recommend that the authors revise this aspect to include appropriate comparisons of SOC and/or respiration rates, wherever available. A further analysis detailing the benefits of this study over previous ones, particularly regarding how depolymerisation rates are influenced by temperature and soil moisture, is advised.

We respond to this feedback in several smaller sections, where we break down and quote the reviewer's points in italics:

*1) Improvements or drawbacks of the model relative to earlier versions and observations:*

We clarify that the JSM model has been previously published and calibrated (Yu et al. 2020; Ahrens et al. 2015, 2020) and refer to these studies often throughout the manuscript. This published model version represents the 'ambient model run' in our manuscript and is used to evaluate the impacts of the different moisture and temperature sensitivities of microbial depolymerisation rates on the SOC pools. We have made this aspect more clear in the manuscript by changing the title of Section 2.4 to "*ambient model run and model experiments*" and including additional information in L. 248 – 250:
"*This ambient model run represents the published C-cycle version of JSM with its default settings (Yu et al. 2020), so that the results of the model experiments with varying temperature and moisture interactions can be evaluated against this default.*"

JSM belongs to the new generation of models that explore SOM decomposition with explicit representations of microbial controls on decomposition and stabilisation of SOM through interactions with mineral surfaces, rather than using empirical representations of SOM turnover. The equations and parameter values in JSM are based on our current scientific understanding and insights from data-driven studies as much as possible and we cite these studies in the Methods section and Table 1. To our knowledge, no other model incorporates distinct temperature and moisture sensitivities for the different SOC decomposition processes. This in itself makes our study with JSM extremely novel and beneficial over previous ones. With JSM, we individually assign and study these intrinsic temperature and moisture sensitivities and their effects on SOC decomposition under different climate change scenarios.

2a) *Further analysis detailing the benefits of this study over previous ones, particularly regarding how depolymerisation rates are influenced by temperature and soil moisture:*

The temperature sensitivity of Vmax has been successfully implemented in JSM before by Ahrens et al. (2020), but a completely novel aspect of our study is the exploration of the individual temperature and soil moisture sensitivities of the depolymerisation rates through the half-saturation constant for depolymerisation ($Km_X$ in Eq. 1). In the manuscript we thoroughly evaluate the temperature sensitivity of Km against the already published model implementation (Yu et al., 2020, ambient model run) which is equal to $Q_{10,Km}$ = 1.0. For these model experiments, the only newly added parameters in our manuscript are $Q_{10,Km,P}$ and $Q_{10,Km,R}$, (Table 1, Table 2), which represent the temperature sensitivities of the half-saturation constants for the depolymerisation of the polymeric litter pool and microbial residues pools, respectively. Our choice for these two parameter values is, like all other parameters in JSM, process-based as much as possible and therefore taken from literature.

In this case, the values are based on a recent laboratory study using enzyme assays by Steven Allison et al. (2018).

2b) *Further analysis detailing the benefits of this study over previous ones, particularly regarding how depolymerisation rates are influenced by temperature and* *soil moisture:*

The moisture dependency of Km was implemented into JSM by Yu et al (2020), but has not been evaluated extensively (Ahrens et al., 2015, 2020; Yu et al. 2020, 2023). We evaluate the effects of this moisture sensitivity against this previously developed version (ambient model run) in the drought experiments, where the Q10 value of Km is kept at 1 (no sensitivity). We show that these moisture effects through Km are particularly important in the subsoil, when microbial biomass is low and effects of mineral sorption are strong, which is important for future model developments that consider microbial dynamics, microbial interactions, and vertically explicit SOC pools.

*3) Include appropriate comparisons of SOC and/or respiration rates, wherever available:*
We would like to reiterate from the previous response, that comparing the model results to measured C content or respiration rates can only be done for past simulations (i.e. the ambient model run), and that such a comparison would not help interpret the results from the perturbation experiments. We clarify that our study serves as a testbed for the various model experiments regarding the intrinsic and apparent temperature and moisture sensitivities of microbial depolymerisation rates. A model like JSM, however, needs climate and litter inputs for its simulation, and instead of generating a synthetic input dataset, we chose Hainich forest because it is a representative example for a forested site on mineral soil in temperate climate. We revised the manuscript to better clarify this (L. 194 – 196): "*JSM requires depth-specific soil temperature, soil moisture and litterfall forcing data at a half hourly time step as input. Following Thum et al. (2019), these soil forcing data were generated for a temperate forest site in Germany as a realistic testbed for in-silico model experiments (Hainich, DE-Hai)*".

Additionally, we discuss and compare the outcomes of our model results against other relevant studies throughout Section 4. Specifically, we discuss the 1) Warming experiments, by comparing our results to other modelling studies (L. 400 – 401) and field studies (L. 401 – 403); and 2) Drought experiments, by comparing our results to other modelling studies (L. 471 – 473) and the very few existing field studies (L.489-501); and 3) Warming & drought experiments, by comparing our results to other modeling studies (L. 510 – 514).

Lastly, we present an explorative modeling study, which showcases the potential impacts of soil warming and drought on SOC stocks along a vertical soil profile. Along this vertical gradient, the interplay between C substrate supply, microbial biomass and mineral sorption changes, which results in different responses to soil warming and drought between the top- and deeper surface layers. As such, we do not report bulk soil fluxes and SOC stock changes in the manuscript, but the relative changes of the different SOC pools at different depths (as % change per year since the start of the experiment) in response to the experimental soil moisture and temperature perturbations. This normalisation was done deliberately, as it allows us to independently compare the effects of the temperature and soil moisture interactions between the model runs.

- The decision to disable the vegetation response to moisture and temperature is questionable. The response of vegetation influences the type, decomposability, and quantity of plant material contributed to the soil, which in turn affects SOC input.

JSM is a stand-alone model, where the litter inputs are generated beforehand by the land surface model QUINCY (described in Section 2.2). We want to reiterate from the previous response that keeping the litter inputs the same for each model experiment is a deliberate choice for this study, and actually a feature rather than a "bug". We mention this in the manuscript's discussion (L. 501 – 503): "*An advantage of our stand-alone soil model environment with prescribed litter inputs is that it allows us to individually test soil warming and drying effects on long-term SOC stocks, while eliminating the potentially confounding effects from changes in plant productivity.*"

Within the current study, the interactions between temperature, soil moisture, mineral stabilisation mechanisms and microbial biomass are already highly non-linear. Adding another major feedback by also changing plant litter inputs over the 100-year simulation period would hide the isolated effects that soil warming and drought can have on SOC stocks. Furthermore, while it is true that the multitude of feedbacks between plant productivity and thus litter production and soil conditions are important, the processes regulating these feedbacks are not so well known that everybody would agree on their model description. In other words, we would need to make many additional assumptions. Overall, we acknowledge that this would be an interesting and important study, but indeed a large and separate study in itself, and therefore falls outside of the scope of this paper. We do discuss the option for a combined vegetation-soil simulation, i.e., where plant productivity simultaneously changes in response to soil warming and drought, in L. 566 – 572 of the manuscript. The option to repeat this study within a coupled plant-soil model framework will be available in the near future, once JSM is fully coupled with the land surface model QUINCY.

Furthermore, the manuscript still lacks an adequate description of the diffusion and advection processes shown in Figure 2. It is essential to explain how they work concerning soil carbon fluxes following the line 215 in the revised version and further elaboration in the discussion. Moreover, the vertical processes described are inadequate to understand how model includes the interaction between the top and bottom layer of the soil.

Following the suggestion of the reviewer we added the following vertically explicit process descriptions to the methodology in section 2.2 (Vertical process presentations in JSM):
- The buildup of the organic layer and the additional downward advective C flux this creates;
- Bioturbation fluxes;
- Vertical DOC transport from percolation.

Specifically, we add (L. 188 – 192):
"*In JSM, all pools are transported via bioturbation according to the formulation by Jarvis et al. (2010) as a diffusive process whose diffusion coefficient decreases as a function of bulk density. The DOC pool is additionally transported via an advection velocity which represents the water mass flow between soil layers. A special feature of JSM is the vertically continuous modelling of potential organic layers and mineral soil layers via an additional advective transport term that accounts for the accumulation and decomposition of organic matter on*

*top and within the soil profile. This ensures that the buildup of organic matter, for example in the form of an organic layer,  leads to a concurrent decrease in the mineral soil volumetric fraction and thereby the sorption capacity qmax. For a full description of JSM, a reference is made to Yu et al. (2020)."*

We also carefully revised section 4.1 of the discussion to better highlight the role of transport (L. 410 – 415):
"*The increase in the proportion of MAOC is partly driven by the advective transport term that represents the downward displacement of mineral matrix and SOC when organic matter builds up in organic layers but also  within the soil profile. The lower SOC concentrations with increasing soil depth lead to a higher proportion of soil volume occupied by minerals compared to organic matter. The higher mineral soil volume thereby provides higher sorption capacities qmax. Adsorption rates in the subsoil are consequently higher than in the topsoil since qmax is farther from saturation.*"

- The authors' focus solely on the first 50 cm of soil, while noting that the lower layers have not reached equilibrium, is concerning. Stating similar pattern observed at depths of 50-100 cm requires further clarification to ensure credibility.

We believe that there is confusion on the importance of steady state for the study. Steady state in the ambient run and as a starting point for our experiments simply facilitates interpretation: we start from steady state for the experiments and stay at steady state for the ambient run (dark blue line in Figure 3, for details on steady state confirmation see our reply below). If we did not start from steady state conditions, there would be an additional confounding factor to disentangle in each model experiment.

In the lower soil layers, it simply takes more time (and hence significantly more computational resources) than 500 spinup years to reach equilibrium.

- Regarding the assessment of steady-state conditions, the authors should provide supporting documents, such as results from the spin-up phase. The claim that steady-state conditions were achieved for each run is not evident in Figure 3, nor convincing overall. Authors are urged to include supporting documentation, such as report from log files, to substantiate this claim.

We think confusion may have arisen from poor phrasing in lines 240 - 242 of the previous manuscript and we corrected this (L. 251 – 253 and L. 297 – 299, see below). We do not claim that steady state was reached after 100 simulation years *at the end of our model experiments*.

Our steady state claim only applies to the beginning of the experiment so that we start each model experiment at steady state condition, i.e. immediately after the 500-year spinup phase.

The manuscript already contains 'supporting documents, such as results from the spinup phase' as requested by the reviewer: The results from the spinup phase are fully presented in the paper as the ambient model run (dark blue line in Fig. 3), where spinup conditions simply continue for another 100 simulation years. Steady state for the ambient run is clearly

evident from visual inspection by the absence of a slope in the dark blue line, but also by the statistical evidence we provide: we tested with a simple linear regression on the slope of the change in the SOC pools for the ambient run between simulation year 0 and simulation year 100, which showed no significant deviation from 0.

This was tested in R, using the linear model "lm" (y = $a$x + $b$). For completeness we include here the result of the regression test for % SOC stock changes (y) in the ambient run over 100 years (x) between 0 - 50 cm depth, corresponding to the blue line in Fig. 3a:

```
lm(formula = y ~ x)

Residuals:
     Min       1Q   Median       3Q      Max
-0.11457 -0.05263  0.02249  0.05477  0.09216

Coefficients:
              Estimate Std. Error t value Pr(>|t|)
(Intercept)  0.0566782  0.0115585   4.904 3.69e-06 ***
y$x 0.0002990  0.0001997   1.497    0.138
* * *
Signif. codes:  0 '***' 0.001 '**' 0.01 '*' 0.05 '.' 0.1 ' ' 1

Residual standard error: 0.05851 on 99 degrees of freedom
Multiple R-squared:  0.02214,   Adjusted R-squared:  0.01226
F-statistic: 2.241 on 1 and 99 DF,  p-value: 0.1376
```

which shows an extremely low estimate for the slope ($a$, 0.0003 % change in SOC stock per simulation year), which is not significant and thus confirms the slope of the relationship approximates 0. There is a very small but significant bias in the estimated intercept ($b$) of 0.0567%. As such, we tested the linear model again but forcing the intercept $b$ through 0 (y = $a$x + 0): the estimated slope ($a$) would be then 0.00114% SOC per simulation year (significant, P < 2e-16), which effectively means that under ambient conditions, SOC stocks between 0-50 cm would increase by ~1% after 1000 simulation years.

To avoid further confusion we changed:
1) L. 251 – 253 in the methods section of the manuscript to "*To ensure that all model experiments started from steady state conditions, we verify that the SOC pools between 0 - 50 cm depth reached steady state after the 500-year spinup period by applying a simple linear regression on the slope of the change in SOC pools for the ambient model run.*"
2) L. 297 – 299 in the results section of the manuscript to "*The ambient model run was conducted as a reference to compare our model experiments to. To check whether JSM reached steady state after spinup, a linear regression test confirmed that the first 6 soil layers of the ambient model run (0 - 50 cm) are in steady state, as there is no SOC loss or accumulation over the complete simulation period (Fig. 3, dark blue).*"

**Other modifications to the manuscript**
During the revisions we found and corrected a few minor spelling and grammar mistakes, and a small mistake in Eq. 3, which are visible in the tracked changes pdf file. In-text positions of the tables and figures were slightly shifted so they would not span across

multiple pages while making efficient use of page space, but their contents are identical to the previously submitted manuscript.

**References**

Ahrens, B., Braakhekke, M. C., Guggenberger, G., Schrumpf, M., and Reichstein, M.: Contribution of sorption, DOC transport and microbial interactions to the 14C age of a soil organic carbon profile: Insights from a calibrated process model, Soil Biology and Biochemistry, 88, 390–402, https://doi.org/10.1016/j.soilbio.2015.06.008, 2015.

Ahrens, B., Guggenberger, G., Rethemeyer, J., John, S., Marschner, B., Heinze, S., Angst, G., Mueller, C. W., Kögel-Knabner, I., Leuschner, C., Hertel, D., Bachmann, J., Reichstein, M., and Schrumpf, M.: Combination of energy limitation and sorption capacity explains 14C depth gradients, Soil Biology and Biochemistry, 148, 107912, https://doi.org/10.1016/j.soilbio.2020.107912, 2020.

Allison, S. D., Romero-Olivares, A. L., Lu, Y., Taylor, J. W., and Treseder, K. K.: Temperature sensitivities of extracellular enzyme Vmax and Km across thermal environments, Glob Chang Biol, 24, 2884–2897, https://doi.org/10.1111/gcb.14045, 2018.

Berg, A., Sheffield, J., and Milly, P. C. D.: Divergent surface and total soil moisture projections under global warming, Geophysical Research Letters, 44, 236–244, https://doi.org/10.1002/2016GL071921, 2017.

Cook, B. I., Mankin, J. S., Marvel, K., Williams, A. P., Smerdon, J. E., and Anchukaitis, K. J.: Twenty-First Century Drought Projections in the CMIP6 Forcing Scenarios, Earth's Future, 8, e2019EF001461, https://doi.org/10.1029/2019EF001461, 2020.

Hsu, H. and Dirmeyer, P. A.: Uncertainty in Projected Critical Soil Moisture Values in CMIP6 Affects the Interpretation of a More Moisture-Limited World, Earth's Future, 11, e2023EF003511, https://doi.org/10.1029/2023EF003511, 2023.

Soong, J. L., Phillips, C. L., Ledna, C., Koven, C. D., and Torn, M. S.: CMIP5 Models Predict Rapid and Deep Soil Warming Over the 21st Century, Journal of Geophysical Research: Biogeosciences, 125, e2019JG005266, https://doi.org/10.1029/2019JG005266, 2020.

Thum, T., Caldararu, S., Engel, J., Kern, M., Pallandt, M., Schnur, R., Yu, L., and Zaehle, S.: A new model of the coupled carbon, nitrogen, and phosphorus cycles in the terrestrial biosphere (QUINCY v1.0; revision 1996), Geoscientific Model Development, 12, 4781–4802, https://doi.org/10.5194/gmd-12-4781-2019, 2019.

Wang, Y., Mao, J., Hoffman, F. M., Bonfils, C. J. W., Douville, H., Jin, M., Thornton, P. E., Ricciuto, D. M., Shi, X., Chen, H., Wullschleger, S. D., Piao, S., and Dai, Y.: Quantification of human contribution to soil moisture-based terrestrial aridity, Nat Commun, 13, 6848, https://doi.org/10.1038/s41467-022-34071-5, 2022.

Yu, L., Ahrens, B., Wutzler, T., Schrumpf, M., and Zaehle, S.: Jena Soil Model (JSM v1.0; revision 1934): a microbial soil organic carbon model integrated with nitrogen and phosphorus processes, Geoscientific Model Development, 13, 783–803, https://doi.org/10.5194/gmd-13-783-2020, 2020.

Yu, L., Caldararu, S., Ahrens, B., Wutzler, T., Schrumpf, M., Helfenstein, J., Pistocchi, C., and Zaehle, S.: Improved representation of phosphorus exchange on soil mineral surfaces reduces estimates of phosphorus limitation in temperate forest ecosystems, Biogeosciences, 20, 57–73, https://doi.org/10.5194/bg-20-57-2023, 2023.